



# Technical note: How are NH₃ dry deposition estimates affected by combining the LOTOS-EUROS model with IASI-NH₃ satellite observations?

Shelley C. van der Graaf[1], Enrico Dammers[2], Martijn Schaap[3,4], Jan Willem Erisman[1,5]

[1]Cluster Earth and Climate, Department of Earth Sciences, Vrije Universiteit Amsterdam, Amsterdam, 1081 HV, The Netherlands

[2]Environment and Climate Change Canada, Toronto, Ontario, ON M3H 5T4, Canada.

[3]TNO, Climate Air and Sustainability, Utrecht, 3584 CB, The Netherlands

[4]Institute for Meteorology, Free University Berlin, Berlin, 12165, Germany

[5]Louis Bolk Institute, Driebergen, 3972, The Netherlands

Correspondence to: Shelley C. van der Graaf (s.c.vander.graaf@vu.nl)

**Abstract.** Atmospheric levels of reactive nitrogen have substantially increased during the last century resulting in

increased nitrogen deposition to ecosystems, causing harmful effects such as soil acidification, reduction in plant biodiversity and eutrophication in lakes and the ocean. Recent developments in the use of atmospheric remote sensing enabled us to resolve concentration fields of NH₃ with larger spatial coverage and these observations may be used to improve the quantification of NH₃ deposition. In this paper we use a relatively simple, data-driven method to derive dry deposition fluxes and surface concentrations of NH₃ for Europe and for the Netherlands. The aim of this

paper is to determine for the applicability and the limitations of this method for NH₃ using space-born observations of the Infrared Atmospheric Sounding Interferometer (IASI)  and the LOTOS-EUROS atmospheric transport model. The original modelled dry NH₃ deposition flux from LOTOS-EUROS and the flux inferred from IASI are compared to indicate areas with large discrepancies between the two and where potential model improvements are needed. The largest differences in derived dry deposition fluxes occur in large parts of Central Europe, where the satellite-

observed NH₃ concentrations are higher than the modelled ones, and in Switzerland, northern Italy (Po Valley) and southern Turkey, where the modelled NH₃ concentrations are higher than the satellite-observed ones. A sensitivity analysis of 8 model input parameters important for NH₃ dry deposition modelling showed that the IASI-derived dry NH₃ deposition fluxes may vary from ~20% up to ~50% throughout Europe. Variations in the dry deposition velocity used for NH₃ led to the largest deviations in the IASI-derived dry NH₃ deposition flux and should be

focused on in the future. A comparison of NH₃ surface concentrations with in-situ measurements of several established networks (EMEP, MAN and LML) showed no significant, or consistent improvement in the IASI-derived NH₃ surface concentrations compared to the originally modelled NH₃ surface concentrations from LOTOS-EUROS. It is concluded that the IASI-derived NH₃ deposition fluxes do not show a strong improvements compared



to modelled NH$_3$ deposition fluxes and there is future need for better, more robust, methods to derive NH$_3$ dry deposition fluxes.

## 1. Introduction

Reactive nitrogen (N$_r$) emissions have substantially increased during the last century to around four times the pre-industrial levels (Erisman et al., 2008;Fowler et al., 2013). As a result atmospheric deposition of reactive nitrogen to both terrestrial and aquatic ecosystems have also increased (Dentener et al., 2006b). Excessive nitrogen deposition to sensitive ecosystems can cause harming effects such as soil acidification, reduction in plant biodiversity and eutrophication in water bodies (Erisman et al., 2015). One molecule of reactive nitrogen may even contribute to a number of these environmental impacts through different pathways and chemical transportation in the biosphere, the so-called nitrogen cascade (Galloway et al., 2003). Ammonia (NH$_3$) is one form of reactive nitrogen and constitutes an important part of the total amount of N$_r$ emissions. Up to 50% of global reactive nitrogen emissions consist of NH$_3$ (Reis et al., 2009) and therefore significantly contributes to these adverse effects. Atmospheric ammonia is deposited to surfaces by two processes: dry and wet deposition.

Dry deposition may comprise a large part of the total deposition. Earlier modelling studies showed that dry deposition of NH$_x$ even constitutes to over 60% of the total deposition (Dentener et al., 2006a). The modelled fraction of dry deposition, however, ranges hugely depending on the used model. Deposition models in general are known to involve large uncertainties regarding the chemistry behind NH$_4$ formation and the NH$_3$ dry deposition velocities (Dentener et al., 2006a). At the same time, large scale assessment of NH$_3$ dry deposition are hindered by the extremely limited number of dry deposition observations and their sparse distribution in space and time. Measurements of NH$_3$ dry deposition fluxes largely remain experimental and are limited to a few research sites and measurement campaigns of short durations (e.g. (Zoll et al., 2016;Spindler et al., 2001)). These measurements typically are representative for a confined area and a specific ecosystem. Dry deposition has so far been estimated on a regional scale through mainly two methods: geostatistical approaches and atmospheric chemistry models. Geostatistical approaches include geospatial interpolation of, or generating statistical models based on existing in-situ observations (e.g. (Erisman and Draaijers, 1995)). Atmospheric chemistry models use known and modelled inputs (i.a. emissions) to derive dry deposition fluxes (e.g. (Dentener et al., 2006a;Wichink Kruit et al., 2012;Van der Swaluw et al., 2017). Both methods depend strongly on the quality and availability of reliable input information, which is often limited or even absent.

Recent development in the use of atmospheric remote sensing to measure NH$_3$ distributions with large spatial coverage and daily resolution (Van Damme et al., 2014a), allowing us to examine their development in space and time in more detail. Information from satellites can be of help to strengthen our understanding of the complex chain of processes of atmospheric deposition, emissions, dispersion, chemistry, especially when complemented with information from atmospheric chemistry models. Atmospheric chemistry models may for example help to fill-in missing information on NH$_3$ concentrations close to the Earth's surface, arising from low sensitivities of NH$_3$ measuring instruments, or may for instance supplement satellite data with information on diurnal cycles. Nowlan et





al. (2014) estimated surface concentrations and dry deposition of $NO_2$ and $SO_2$ by combining satellite observations of the Ozone Monitoring Instrument (OMI) and the GEOS-Chem model. The resulting estimates compared reasonably well with in-situ measurements, thus providing a relatively simple, data-driven method to estimate surface concentrations and dry deposition fluxes on a world-wide scale. More recently, Kharol et al. (2017) derived

$NH_3$ dry deposition fluxes over North-America using a similar method with $NH_3$ observations of the Cross-track Infrared Sounder (CrIS) satellite and the GEM-MACH model. The aim of this paper is to search for the applicability and the limitations of this method for $NH_3$ over Europe using space-born observations of the Infrared Atmospheric Sounding Interferometer (IASI) and the LOTOS-EUROS atmospheric transport model. This paper shows the first use of the IASI-$NH_3$ product for the derivation of $NH_3$ dry deposition fluxes, together with validation of the derived

$NH_3$ surface concentrations with in-situ measurements. The latter serve as a direct proxy for the validity of the derived $NH_3$ dry deposition fluxes. Also, this paper is the first to estimate the effect of modelling errors on the satellite-derived $NH_3$ dry deposition fluxes by performing a model sensitivity study.

We start this paper with a description of the used models and datasets and their associated uncertainties. This is followed by a description of the methodology that is used to determine the $NH_3$ surface concentrations and dry

deposition fluxes and the design of a sensitivity study of the LOTOS-EUROS model. The resulting estimates of the $NH_3$ surface concentrations and dry deposition fluxes are given and compared to in-situ measurements from the EMEP network throughout Europe, and to in-situ measurements from the LML and the MAN network in a special case-study for the Netherlands. Moreover, a sensitivity study of the LOTOS-EUROS model is performed to estimate the effect of model input uncertainties on the results that are obtained in the same section. The study is then

concluded with a discussion.

## 2. Models and datasets

### 2.1 IASI-$NH_3$ product

The Infrared Atmospheric Sounding Interferometer (IASI) is a passive remote-sensing instrument that measures infrared radiation emitted by the Earth's surface and atmosphere within the spectral range of 645-2769 cm$^{-1}$

(Clerbaux et al., 2009). The IASI-A instrument is on board of the MetOp-A satellite which was launched in 2006 and circles in a polar sun-synchronous orbit. In this study we used $NH_3$ total column measurements from the morning overpass, as these are more sensitive to $NH_3$ then the nighttime observations (Van Damme et al., 2015). The $NH_3$ product has an elliptical spatial footprint of approximately 12 by 12 kilometers and a detection limit of 2.5 ppbv (Van Damme et al., 2015). The retrieval uses a neural network to derive $NH_3$ columns based on the

calculation of the HRI (Hyperspectral Range Index), e.g. the spectral index (Van Damme et al., 2017). This retrieval algorithm combines information on the temperature, humidity and pressure profiles to closely represent the atmospheric state (Whitburn et al., 2016). The retrieval uses a fixed profile in time, based on the profiles described by Van Damme et al. (2015). The IASI-NN retrievals have been validated in Dammers et al. (2016) and Dammers et al. (2017b). In these papers they compared the IASI-NN and FTIR total columns and showed that the two compare

reasonably well with a systematic underestimation by the IASI-NN product of around 30%. In this paper the $NH_3$ total columns observed during the warmer season (April to September) of 2013 and 2014 are used. The warm season


was chosen because considerably fewer observations are available during the cold months. Moreover, the observations in the cold months generally have a higher relative uncertainty (Van Damme, 2014). A filter has been applied after (Van Damme et al., 2014b), filtering out observations with an relative error of <100% unless the absolute error is smaller than 5x $10^{15}$ molecules $cm^{-2}$. Figure 1 shows the mean IASI $NH_3$ total column

concentration over Europe and the Netherlands.

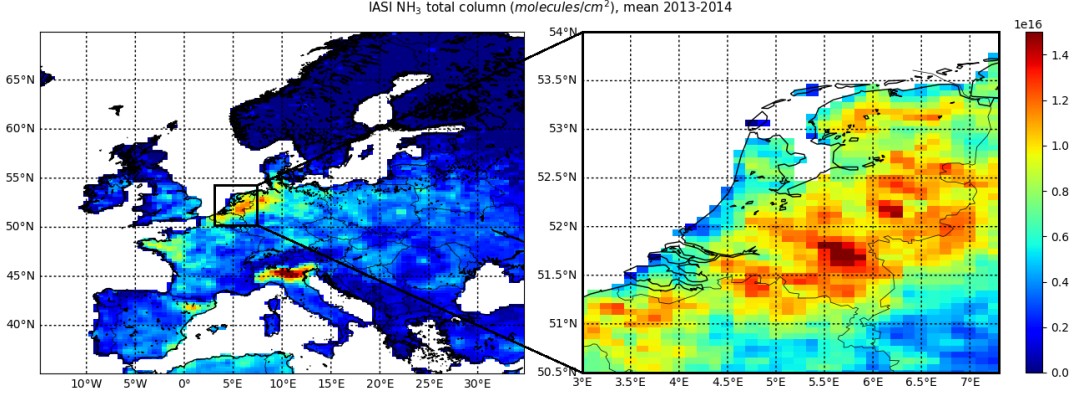

**Figure 1: The annual mean $NH_3$ total column concentration in 2013- 2014 as observed by IASI-A in Europe (regridded to 0.50⁰ longitude by 0.25⁰ latitude) and the Netherlands (regridded to 0.125⁰ longitude by 0.0625⁰ latitude).**

### 2.2  IASI $NH_3$ uncertainties

The retrieval algorithm (Whitburn et al., 2016) allows estimation of a quantitative errors of each observation. The error estimate depends on a combination of the thermal contrast (the temperature difference between Earth's surface and atmosphere at 1.5km) and the HRI, i.e. the spectral footprint, and includes error terms for the uncertainty in the profile shape, and error terms arising from the used temperature and water vapor profiles. More information on the IASI-NN satellite retrieval and how the relative errors are derived can be found in Whitburn et al. (2016). Figure 2

shows the relative uncertainty of the IASI-A $NH_3$ total column concentrations in 2013-2014 over Europe and the Netherlands. The relative uncertainty ranges from ~90% in remote areas with little emissions to ~30% in high emissions areas.



Relative uncertainty (%)

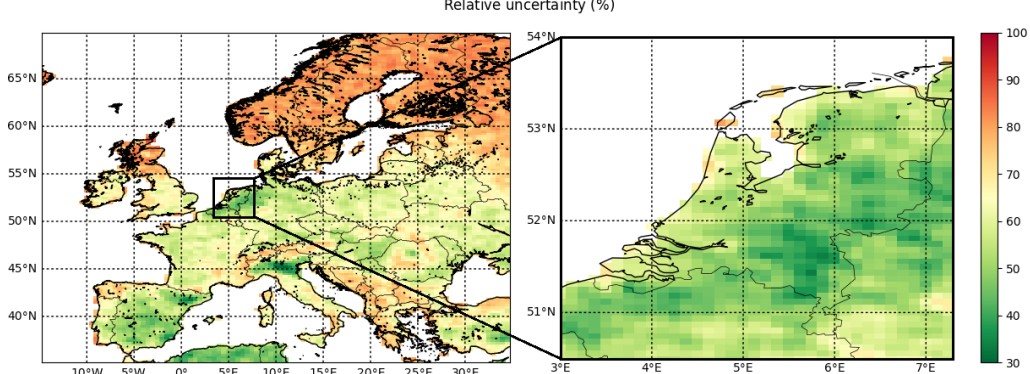

**Figure 2:** **The relative error of the annual IASI-A retrieved NH$_3$ total column concentrations in Europe and the Netherlands in 2013-2014.**

### 2.3  NH$_3$ ground measurements

Ground measurements of NH$_3$ surface concentrations from three air quality networks were used to validate the NH$_3$ surface concentrations from LOTOS-EUROS and the ones derived from IASI, both on a monthly and a yearly basis. To do this, observations of ambient NH$_3$ concentrations of the EMEP (European Monitoring and Evaluation Program) network are used for Europe (EMEP, 2016). For the case study of the Netherlands observations from two established networks are used, the LML network (Landelijk Meetnet Luchtkwaliteit) (RIVM) and the MAN

(Meetnet Ammoniak in Natuurgebieden) (Lolkema et al., 2015).

NH$_3$ is challenging to reliably measure because of potential adsorption to parts of the measurement device, leading to slow response times (von Bobrutzki et al., 2010). The uncertainties of the measurements may differ significantly per instruments design. Table 1 gives an overview of the instruments used by each of these networks and their uncertainties.

#### 2.3.1  EMEP network

The main measurement network for reactive nitrogen concentrations on European-scale is the EMEP (European Monitoring and Evaluation Program) network (Tørseth et al., 2012). NH$_3$ measurements from 35 stations were used to validate the results of 2013 and 46 stations were used for 2014. Different types of measurement devices are used to measure NH$_3$ within the EMEP network. The majority of the EMEP sites use filter-packs, of which the results are

relatively uncertain. In a field intercomparison of different NH$_3$ measurement techniques (von Bobrutzki et al., 2010) showed that different instruments have an overall bias varying from -31.1% to +10.9% for the entire data range (~two weeks), demonstrating that there is a need for standardized approach. For smaller concentrations (<10 ppbv) the bias is even larger, from -22.0% to +54.5%.

#### 2.3.2  LML network

The LML monitors hourly NH$_3$ concentrations in the Netherlands since 1993 (van Zanten et al., 2017). Since 2014




only six stations are left in operation, before that there were eight stations. The locations of the monitoring stations were carefully selected to equally cover regions with high, moderate and low emission densities. The measurements are performed with AMOR instruments, which are continuous flow denuders. An airflow passes through a wetted rotating denuder tube in the AMOR instrument and the $NH_3$ absorbs into this fluid. The electric conductivity is then

determined and used as a measure for the $NH_3$ concentration (van Zanten et al., 2017). The measurements have a reported uncertainty of at least 9% for hourly concentrations and at least 7% for yearly averages (van Zanten et al., 2017;Blank, 2001).

### 2.3.3   MAN network

The MAN network provides monthly mean ambient $NH_3$ concentrations in nature areas in the Netherlands since

2005. The network has a total of 236 sampling points since 2014, spread over 60 different nature areas. The measurements are performed with low-cost passive samplers from Gradko that are calibrated against the measurements of the LML. The bottom of the passive sampler is an open cap with a porous filter through which $NH_3$ in air can enter. In the top end of the tube the $NH_3$ is adsorbed by an acid to form $NH_4^+$. The $NH_4^+$ concentrations in the samplers are analyzed in a laboratory every month to compute the monthly mean $NH_3$

concentrations. The uncertainty of the MAN measurements depends on the $NH_3$ concentration and varies between 20% for high concentrations (10-20 $\mu gm^{-3}$) and 41% for low concentrations (1 $\mu gm^{-3}$) (Lolkema et al., 2015).

| Network | Instrument(s) | Uncertainty |
|---------|---------------|-------------|
| EMEP | Filter-packs, denuders | ~20 – 25 % (yearly means) |
| MAN | Passive samplers | 20 – 41% (monthly means) (Lolkema et al., 2015) |
| LML | Continuous-flow denuders (AMORs) | > 9% (hourly measurement), > 7% (observed annual means) (Blank, 2001) |

**Table 1: Type of instruments used to measure ambient $NH_3$ concentrations and associated uncertainty estimates.**

### 2.4   The LOTOS-EUROS model

#### 2.4.1   Model description

LOTOS-EUROS is an Eulerian chemistry transport model (CTM) (Manders et al., 2017) that simulates air pollution in the lower troposphere. For this study a horizontal resolution of 0.50º longitude by 0.25º latitude, corresponding to approximately 28 by 28 $km^2$ is used to perform simulations for Europe (35ºN - 70ºN, 15ºW -35ºE). Secondly, for the case study of the Netherlands the horizontal resolution is set to 0.125º longitude by 0.0625º latitude, approximately 7 by 7 km (50.5 ºN - 54ºN, 3ºE -7.5ºE). The vertical resolution of the model is a four-layer vertical grid that extends

up to 3.5 km above sea level. The bottom layer is the surface layer and has a fixed height of 25 meters. On top of this layer there is a mixing layer, followed by two equally thick dynamic reservoir layers with time-varying thicknesses. The model follows the mixed layer approach. LOTOS-EUROS performs hourly calculation using meteorology provided by ECMWF (ECMWF, 2016). Gas-phase chemistry is described using the TNO CBM-IV scheme (Schaap et al., 2009), which is an updated version of the original scheme by (Whitten et al., 1980).



Anthropogenic emissions used in LOTOS-EUROS are taken from the TNO-MACC-III emission database (Kuenen et al., 2014).

### 2.4.2 Dry deposition parameterization

The dry deposition fluxes in LOTOS-EUROS are calculated with the DEPAC3.11 (Deposition of Acidifying

Compounds) module, following the resistance approach (van Zanten et al., 2010). In this approach, the deposition velocity is the reciprocal sum of the aerodynamic resistance, the quasi-laminar layer resistance and the surface resistance. A canopy compensation point for simulation of the bi-directional flux of $NH_3$ is included in the implementation of the DEPAC3.11 module, following the approach presented in Wichink Kruit et al. (2012). The compensation point is dynamically computed using modelling results from the last month. The model uses the

CORINE/Smiatek land use map converted to the DEPAC land use classes to determine the exchange velocities for different land use classes. More information on the LOTOS-EUROS model can be found in Manders et al. (2017).

### 2.4.3 Model performance

The LOTOS-EUROS model has participated in multiple model intercomparison studies (e.g. (Colette et al., 2017;Wichink Kruit, 2013;Bessagnet et al., 2016)), showing an overall good model performance. LOTOS-EUROS

also showed a good correspondence with yearly $NH_3$ concentrations with a slight underestimation in agricultural areas and overestimation in nature areas in the Netherlands (Wichink Kruit, 2013).

The inferential method that we use here heavily relies on results from LOTOS-EUROS and therefore, if we wish to obtain reasonable results, the model has to closely represent reality. As in any model, there are, however, uncertainties associated with every part of the total chain of modelled processes. The uncertainties related to

emissions and to dry and wet deposition are expected to show the largest impact on the results and are therefore discussed below.

### 2.4.4 Uncertainties related to emission input

Emissions are the most important input for the any CTM and are, at the same time, a source of substantial uncertainties (Reis et al., 2009;Behera et al., 2013). $NH_3$ emissions are relatively uncertain due to the diverse nature

of agricultural sources leading to large spatial and temporal variations in emissions. The uncertainty of the European reported annual totals is estimated to be around ±30% (EMEP, 2016). The uncertainty is larger for countries that have limited research on their emission inventory and carry out few emission measurement activities.

The presence of other gaseous components such as $SO_2$ and $NO_x$ may have a high impact on the modelled $NH_3$ concentrations, as $NH_3$ in the atmosphere reacts readily with sulfuric acid ($H_2SO_4$) and nitric acid ($HNO_3$) to form

particulate ammonium (e.g. $(NH_4)_2SO_4$ or $NH_4NO_3$). It is therefore also important to consider the errors in the $SO_2$ and $NO_x$ emissions. The $SO_2$ emissions are relatively well known per source category and thus hold a relatively low uncertainty of about ±10% on reported annual totals. The uncertainty in the $NO_x$ emissions is higher, of around ±20% on reported annual totals. However, due to interpolation to account for missing data for some countries, the final uncertainty of the annual totals of both $SO_2$ and $NO_x$ is estimated to be higher (Kuenen et al., 2014).



Needless to say, one single emission at a certain time may have a much higher error due to the large uncertainty related to redistribution and the timing of emissions (Hendriks et al., 2016;Skjøth et al., 2011). More information on the quality data ratings of $NH_3$, $SO_2$ and $NO_x$ per source category and per country can be found in the report of the European Environment Agency (EEA, 2016).

### 2.4.5    Uncertainties regarding dry and wet deposition

The second source of uncertainties originates from the model parameterization of both dry and wet deposition. Several multi-model studies (e.g. (Dentener et al., 2006a;Colette et al., 2017;Wichink Kruit, 2013;Flechard et al., 2011) have shown that there is quite a large discrepancy in the implementation of dry and wet deposition in different CTMs. A fundamental input for estimating dry deposition fluxes in CTMs is uncertainty in the deposition velocity.

Schrader and Brummer (2014) compiled a database of the $NH_3$ deposition velocities per land use category that are used in several deposition models from 2004 to 2013. The results showed that there is quite a large variation in the $V_d$ values that are used for different land use classes. Some classes (e.g. water, urban) showed only a small variation in $V_d$ of an interquartile range of ~5 to 10% for 50% of the data, whereas other classes (e.g. coniferous, agriculture) showed a much larger interquartile range in $V_d$ of ~30 to 40%. Flechard et al. (2011) compared four existing dry

deposition routines across 55 $N_r$ monitoring sites and found that the differences between models reach a factor 2-3 and are often larger than differences between monitoring sites. (Erisman, 1993) estimated the dry and wet deposition fluxes of acidifying substances in the Netherland from measured and modelled concentrations. The estimated uncertainty in the average $NH_3$ fluxes in this paper was estimated to be 30%, with a systematic error of 30% in the used $V_d$ for $NH_3$. Dentener et al. (2006a) calculated the deposition of $N_r$ with 23 atmospheric chemistry transport

models in a multi-model evaluation. Although there were quite large differences between the different models, the paper showed that 71.7% of the model-calculated mean wet deposition rates in Europe agreed to within ±50% with $NH_4^+$ wet deposition measurements from the EMEP network.

## 3    Methodology

The $NH_3$ surface concentrations and the dry deposition fluxes are estimated by combining the observations of the

IASI-A satellite instrument and the modelling results from LOTOS-EUROS, following the  approach for $NO_2$ and $SO_2$ presented by Nowlan et al. (2014). The daytime overpass of the IASI-A satellite instrument passes over Europe once a day in the morning at around 9:30, and thus we only have measurements of the $NH_3$ total column concentration at this specific time. The LOTOS-EUROS model results are used to account for the diurnal variation in $NH_3$ atmospheric concentrations in the computation of both the $NH_3$ surface concentrations and $NH_3$ dry

deposition flux. Moreover, the vertical $NH_3$ profiles in LOTOS-EUROS are used to deduce information about the ground-level $NH_3$ concentrations.

### 3.1 Surface concentration computation

To derive the monthly mean $NH_3$ surface concentrations for Europe, the observed IASI $NH_3$ total column concentrations are first regridded onto the LOTOS-EUROS model grid. The monthly mean $NH_3$  total column



concentrations are then calculated for each pixel. The satellite-derived $NH_3$ surface concentrations $C^{IASI}$ are computed per grid cell using the modelled results from LOTOS-EUROS in the following way Eq. (1):

$$C^{IASI} = \frac{\Omega^{IASI}}{\Omega^{LE}_{overpass}} \cdot C^{LE} \qquad (1)$$

Here $\Omega^{IASI}$ represents the monthly mean IASI $NH_3$ total column concentration (molecules cm$^{-2}$), $\Omega^{LE}_{overpass}$ represents the modelled monthly mean $NH_3$ total column concentration at overpass time (molecules cm$^{-2}$) and $C^{LE}$ is the modelled mean surface concentration (μg m$^{-3}$), the concentration in the down-most layer in LOTOS-EUROS.

### 3.2 Dry deposition flux computation

The computation of the $NH_3$ dry deposition flux is adapted after the approach used by Nowlan et al. (2014), who estimated the $NO_2$ and $SO_2$ dry deposition fluxes using space-born measurements from the Ozone Monitoring

Instrument (OMI) and the GEOS-Chem model. We directly use the vertical profile of $NH_3$ per grid cell in LOTOS-EUROS to relate the IASI $NH_3$ total column to $NH_3$ surface concentrations. The $NH_3$ dry deposition flux (kg N ha$^{-1}$yr$^{-1}$) inferred from IASI, $F^{IASI}$, is therefore computed as follows Eq. (2):

$$F^{IASI} = \frac{\Omega^{IASI}}{\Omega^{LE}_{0overpass}} \cdot F^{LE}_{daily} \qquad (2)$$

Here $\Omega^{IASI}$ denotes the $NH_3$ total column concentration from IASI, $\Omega^{LE}_{0overpass}$ the simulated $NH_3$ total column at

overpass time in LOTOS-EUROS (molecules cm$^{-2}$) and $F^{LE}_{daily}$ the total daily $NH_3$ dry deposition flux in LOTOS-EUROS (kg N ha$^{-1}$yr$^{-1}$), which is the sum of the hourly $NH_3$ dry deposition fluxes Eq. (3):

$$F^{LE}_{daily} = \sum_{h=1}^{24} F^{LE}_h = \sum_{h=1}^{24} V_d \left( C^{LE}_h - \chi^{LE}_{tot,h} \right) \qquad (3)$$

The hourly $NH_3$ dry deposition fluxes is the product of the dry deposition velocity $V_d$ and the difference between the hourly $NH_3$ surface concentration, $C^{LE}_h$, and the total compensation point of $NH_3$, $\chi^{LE}_{tot,h}$. To account for the high

variability of atmospheric $NH_3$ and the limiting amount of available IASI observations monthly means of these values are used rather than daily values.

### 3.3 Sensitivity analysis

The main sources of model uncertainties that are relevant for deposition modelling arise from uncertainties in the emission input and the deposition parameterizations (see Section 2.3).

A total of four input fields were varied in LOTOS-EUROS: the MACC-III $NH_3$ emissions, the MACC-III $NO_x$ and $SO_2$ emissions, the dry deposition velocity, $V_d$, of $NH_3$ and the wet deposition of $NH_3$ by adjusting the used gas scavenging constant, $G_{scav}$, for $NH_3$. The wet scavenging constant $G_{scav}$ linearly influences the amount of wet deposition of $NH_3$ resulting in changes in the wet $NH_3$ deposition flux of +30% and -30%, too. The objective of these 8 sensitivity runs is to assess the uncertainty ranges on the estimated dry $NH_3$ deposition fluxes resulting from



modelling errors. Table 2 gives an overview of the parameters that are varied. We chose to apply a constant perturbation of +30% and -30% to one field at the time to see their individual effect and also to improve the comparability of the results. Moreover, perturbations of ±30% are reasonable ranges since they correspond to the estimated uncertainties in the MACC-III emission fields annual totals and the uncertainties in the wet and dry

deposition fluxes of $NH_3$.

| Perturbed parameter | Perturbations |
|---|---|
| MACC-III $NH_3$ emissions | +30%, -30% |
| MACC-III $NO_x$ and $SO_2$ emissions | +30%, -30% |
| $NH_3$ dry deposition velocity, $V_d^{NH3}$ | +30%, -30% |
| $NH_3$ gas scavenging coefficient, $G_{scav}^{NH3}$ | +30%, -30% |

**Table 2: Perturbations on input fields that have been used for the sensitivity analysis of the method.**

## 4    Results

### 4.1    $NH_3$ surface concentrations

#### 4.1.1    Europe

Figure 3 shows the warm season (April-September) mean $NH_3$ surface concentrations modelled in LOTOS-EUROS and inferred from IASI and the coinciding warm season mean $NH_3$ surface concentrations measured by the EMEP stations in 2013 and 2014. In general, the pattern of the EMEP $NH_3$ surface concentrations and the $NH_3$ surface concentrations from LOTOS-EUROS and IASI matches quite well. The majority of the EMEP stations agree with

the mean $NH_3$ surface concentration from LOTOS-EUROS and inferred from IASI to -0.75 to +0.75 $\mu g m^{-3}$. The sum of the absolute differences between the warm season mean $NH_3$ surface concentrations in a cubic meter from EMEP and LOTOS-EUROS was 23.0 $\mu g$ in 2013 and 32.5 $\mu g$ in 2014. The sum of the absolute differences between the warm season mean $NH_3$ surface concentrations from EMEP and IASI was slightly lower, 22.6 $\mu g$ in 2013 and 28.0 $\mu g$ in 2014.



**Figure 3: Comparison of the warm season (April-September) mean NH$_3$ surface concentrations (µgm$^{-3}$) from LOTOS-EUROS and derived from IASI and the warm season mean NH$_3$ surface concentrations measured by the EMEP stations in 2013 (a, b, c, d) and 2014 (e, f, g, h). The differences between the yearly mean NH$_3$ surface concentrations are shown in the right figures.**



Figure 4 shows a scatterplot of the monthly mean $NH_3$ surface concentrations measured by the EMEP stations compared to the monthly mean $NH_3$ surface concentrations from LOTOS-EUROS and derived from IASI. The LOTOS-EUROS and the EMEP network monthly mean $NH_3$ surface concentration show a reasonably strong linear relationship in 2013 (r =0.71). In 2014 the correlation between the two was weaker (r = 0.39). The correlation

between the IASI-derived versus the EMEP $NH_3$ surface concentrations shows similar correlations in 2013 (r = 0.71), but was higher in 2014 (r = 0.46). The comparison of the warm season mean $NH_3$ concentrations in both 2013 and 2014 shows that the IASI-derived $NH_3$ surface concentrations have a slightly improved correlation coefficient and slope compared to the originally modelled $NH_3$ surface concentrations from LOTOS-EUROS.

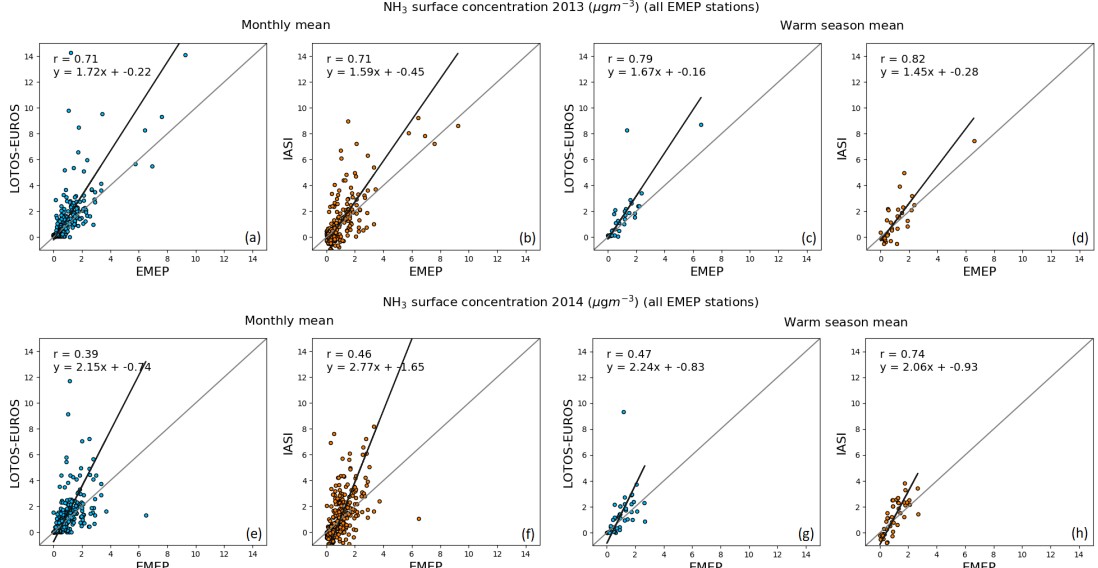

**Figure 4: Comparison of the monthly mean (a, b, e, f) and warm season (April-September) mean (c, d, g, h) NH$_3$ surface concentrations measured by the EMEP stations and the corresponding NH$_3$ surface concentrations from LOTOS-EUROS (blue dots) and inferred from IASI (orange dots) in 2013 (top) and 2014 (bottom).**

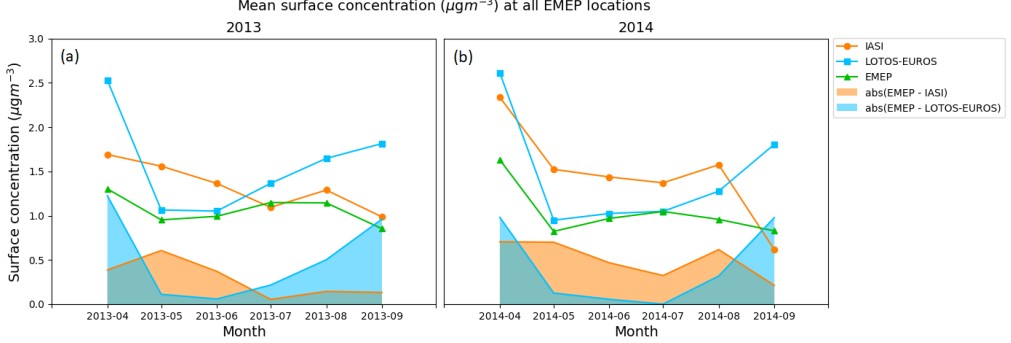

**Figure 5: Mean of the NH$_3$ surface concentrations at all EMEP locations per month (green line), and the coinciding NH$_3$**

**surface concentrations from LOTOS-EUROS (blue line) and IASI (orange line) in 2013 (a) and 2014 (b). The absolute**



**differences between EMEP and LOTOS-EUROS are shown in blue and the absolute differences between EMEP and IASI
are shown in orange.**

Figure 5 shows the mean $NH_3$ surface concentration of all EMEP stations per month, and the corresponding mean
$NH_3$ surface concentration from LOTOS-EUROS and IASI at the same locations. The absolute differences per

month are plotted in the same figure in blue (LOTOS-EUROS vs EMEP) and orange (IASI-derived vs EMEP). All
concentration time profiles show a peak value in April, resulting from spring fertilization. The LOTOS-EUROS time
profile at the EMEP locations in both 2013 and 2014 decreases from April to May and starts to increase towards the
end of the year. The time profile of the mean $NH_3$ surface concentration of the EMEP stations follows this pattern
from April to June, but decreases towards the end of the year. The IASI-derived time profile shows a decreasing

pattern, except in August, where there is a small peak. The IASI-derived time profile shows a relatively better
comparison with the EMEP measurements in April and July to September in 2013 and in April and September in
2014. The sum of the absolute differences of the mean $NH_3$ surface concentrations in a cubic meter at all EMEP
locations between LOTOS-EUROS and EMEP amounts to 3.1 µg in 2013 and 2.5 µg in 2014. The sum of the
absolute differences between IASI and EMEP was somewhat smaller in 2013, amounting to 1.7 µg, and somewhat

higher in 2014, amounting to 3.0 µg.

In summary, there appears to be some minor improvements in the IASI-derived $NH_3$ surface concentrations
compared to the modelled $NH_3$ surface concentrations from LOTOS-EUROS on a monthly basis. The comparison
of warm season means shows that there is a more pronounced improvement in the IASI-derived $NH_3$ surface
concentrations compared to the modelled $NH_3$ surface concentrations from LOTOS-EUROS on a seasonal basis.

**4.1.2    The Netherlands**

**4.1.2.1   Comparison with LML measurements**
Figure 6 shows the LOTOS-EUROS and IASI-derived warm season (April-September) mean $NH_3$ surface
concentrations ($\mu gm^{-3}$) in the Netherlands in 2013 and 2014 and the coinciding LML observations. LOTOS-EUROS
seems to capture the general pattern in $NH_3$ surface concentrations fairly well in both 2013 and 2014.  The sum of

the absolute differences between the warm season mean $NH_3$ surface concentrations in a cubic meter from LML and
LOTOS-EUROS was 47.3 µg in 2013 and 44.8 µg in 2014. The sum of the absolute differences between the warm
season mean $NH_3$ surface concentrations from LML and IASI was slightly lower in 2013, namely 44.9 µg, and
somewhat higher in 2014, namely 48.5 µg.





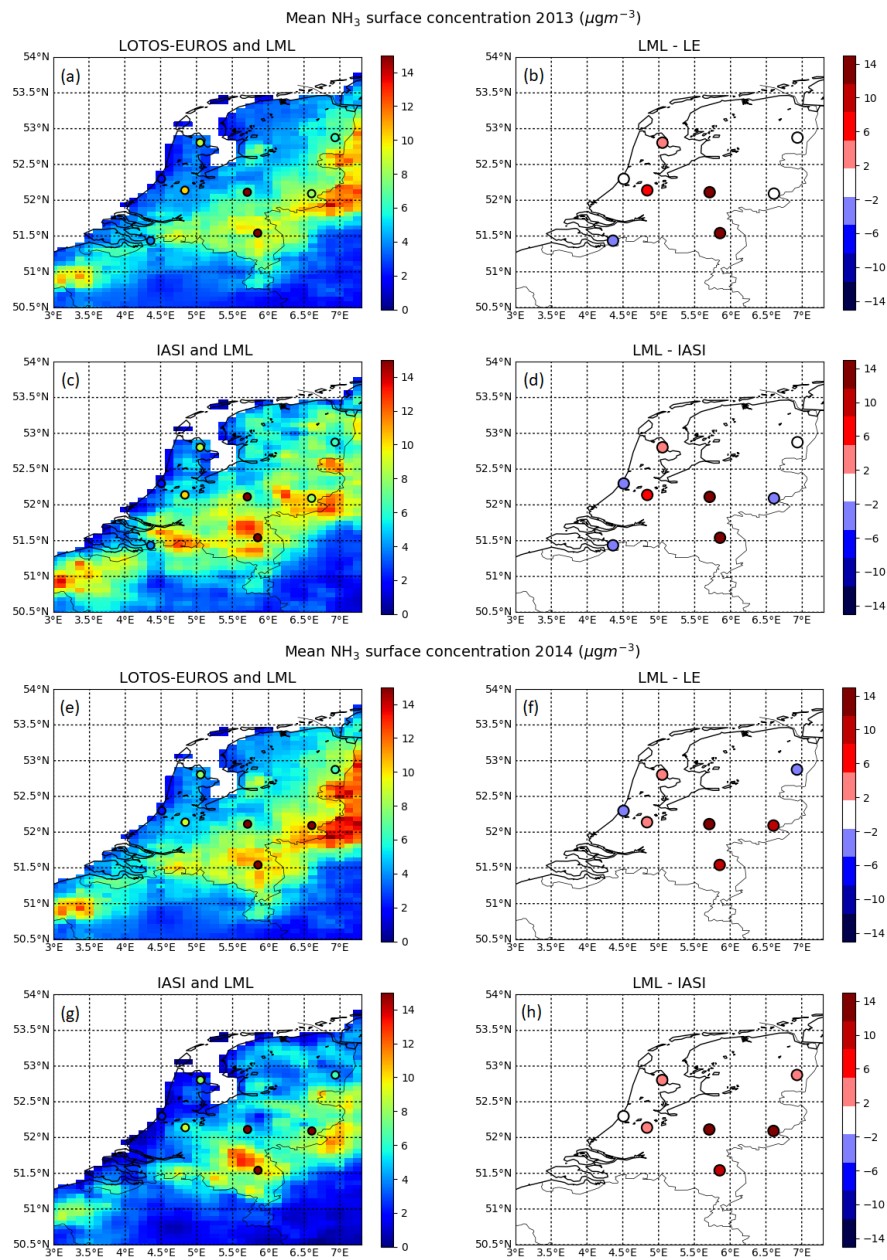

**Figure 6: Comparison of the warm season (April-September) mean NH₃ surface concentration in 2013 (a, b, c, d) and in 2014 (e, f, g, h) from LOTOS-EUROS and derived using IASI. The corresponding warm season mean NH₃ surface concentrations measured by the LML stations are plotted on top of the left figures. The right figures depict the differences between the two.**



Figure 7 shows a scatterplot of all LML monthly mean surface concentrations ($\mu gm^{-3}$) versus the corresponding LOTOS-EUROS and IASI-derived concentrations. The LOTOS-EUROS and the LML monthly mean NH$_3$ surface concentration show a moderate linear relationship (r =0.39 in 2013, r = 0.50 in 2014). The high NH$_3$ concentration level stations Vredepeel and Wekerom are underestimated by LOTOS-EUROS, the other stations are closer to the

one-on-one line and appear to match quite well. The correlation coefficient of the IASI-derived and the LML concentrations was r =0.39 in 2013 and r = 0.53 in 2014. The IASI-derived surface concentrations also underestimate the high concentration LML stations (Vredepeel and Wekerom) both in 2013 and 2014. The majority of the low-concentration LML stations are overestimated by the IASI-derived concentrations in 2013, and underestimated by the IASI-derived concentrations in 2014. In general, both high and low LML NH$_3$ surface

concentrations were inadequately reproduced by the IASI-derived NH$_3$ surface concentrations. The exclusion of the high-concentration level stations Vredepeel and Wekerom did not lead to a better comparison of the LML and the IASI-derived NH$_3$ surface concentrations.

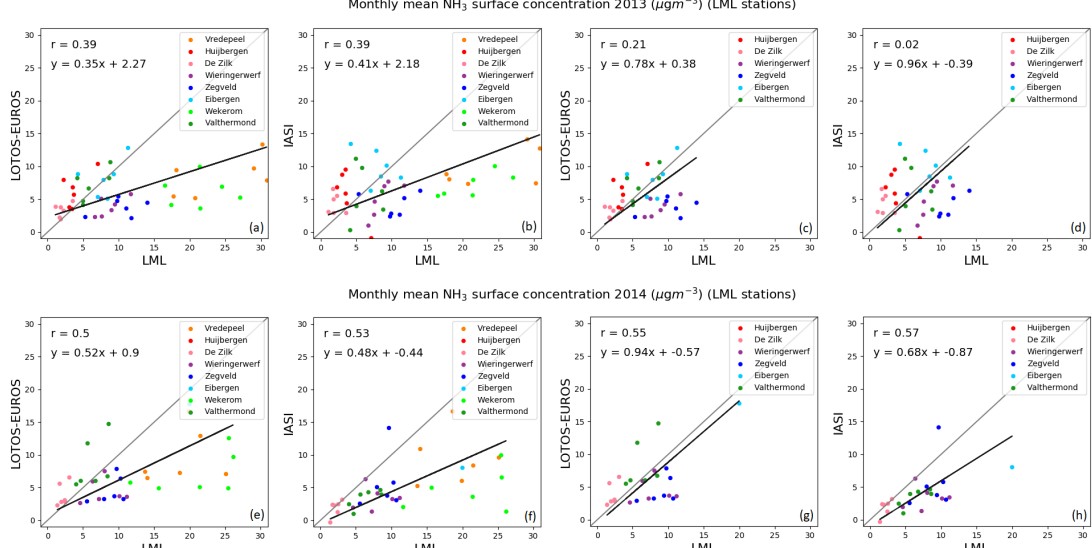

**Figure 7: Comparison of the monthly mean NH$_3$ surface concentrations measured by the LML stations and the**

**corresponding LOTOS-EUROS and IASI-derived NH$_3$ surface concentrations during the warm season (April-September) of 2013 (top) and 2014 (bottom). The high-concentration stations Vredepeel and Wekerom are excluded from the right figures (c, d, g, h).**

Table 3 gives a month by month comparison of the correlation coefficient r, the slope and the intercept of the monthly mean NH$_3$ surface concentrations of all LML stations versus the corresponding LOTOS-EUROS and IASI-

derived surface concentrations. In 5 out of 12 months the correlation coefficient and the slope of the IASI-derived surface concentrations have been improved compared to the LOTOS-EUROS surface concentrations, mainly during the summer months. The improvements in 2013 could potentially be related to the fact that the NH$_3$ total columns from IASI were higher than the modelled NH$_3$ total columns and that LOTOS-EUROS underestimated the majority of the LML NH$_3$ surface concentrations.



In summary, the comparison with the LML stations does not show any significant, or consistent improvement in the IASI-derived NH$_3$ surface concentrations compared to the originally modelled NH$_3$ surface concentrations from LOTOS-EUROS.

| | month | LOTOS-EUROS | | | | IASI-derived | | | |
|---|---|---|---|---|---|---|---|---|---|
| | | r | slope | intercept | RMSD | r | slope | intercept | RMSD |
| | 04-2013 | 0.57 | 0.39 ↑ | 4.12 | 7.78 ↑ | 0.57 | 0.36 | 0.01 ↑ | 10.80 |
| | 05-2013 | 0.49 ↑ | 0.19 ↑ | 2.16 ↑ | 7.53 | -0.21 | -0.30 | 9.61 | 7.20 ↑ |
| | 06-2013 | 0.38 | 0.19 | 1.73 ↑ | 8.58 | 0.44 ↑ | 0.45 ↑ | 1.74 | 6.80 ↑ |
| | 07-2013 | 0.36 | 0.18 | 3.31 ↑ | 11.67 | 0.46 ↑ | 0.34 ↑ | 3.74 | 10.00 ↑ |
| | 08-2013 | 0.49 | 0.23 | 3.82 | 10.10 | 0.86 ↑ | 0.35 ↑ | 3.63 ↑ | 7.93 ↑ |
| | 09-2013 | 0.27 ↑ | 0.33 | 4.28 | 5.79 ↑ | 0.04 | 0.65 ↑ | 0.38 ↑ | 7.31 |
| LML | 04-2014 | 0.69 ↑ | 0.56 ↑ | 4.36 | 5.81 ↑ | 0.21 | 0.46 | 0.44 ↑ | 10.32 |
| | 05-2014 | 0.39 | 0.29 | 1.90 ↑ | 6.35 | 0.76 ↑ | 0.72 ↑ | -2.79 | 6.15 ↑ |
| | 06-2014 | 0.63 | 0.20 | 2.31 | 9.65 | 0.85 ↑ | 0.66 ↑ | -0.99 ↑ | 6.60 ↑ |
| | 07-2014 | 0.70 ↑ | 0.19 | 2.27 | 10.53 | 0.68 | 0.29 ↑ | 1.22 ↑ | 10.19 ↑ |
| | 08-2014 | 0.68 ↑ | 0.47 ↑ | 0.75 | 4.97 ↑ | 0.46 | 0.31 | 0.69 ↑ | 6.50 |
| | 09-2014 | 0.55 ↑ | 0.33 ↑ | 4.84 | 8.20 ↑ | 0.04 | 0.27 | 1.49 ↑ | 11.59 |

**Table 3: Month by month comparison of the correlation coefficient (r), slope and intercept of the monthly mean NH$_3$ surface concentrations of the LML stations (x-axis) and the coinciding monthly mean LOTOS-EUROS and IASI-derived NH$_3$ surface concentrations (y-axis). The green arrows denote which of the two (LOTOS-EUROS or IASI) gives the most desirable values. The green arrows are attributed to either LOTOS-EUROS or IASI based on the following criteria: highest r, slope closest to 1, intercept closest to 0 and smallest RMSD.**

### 4.1.2.2 Comparison with MAN measurements

Figure 8 shows the comparison of the LOTOS-EUROS and IASI-derived and the MAN warm season mean NH$_3$ surface concentrations. The LOTOS-EUROS model captures the pattern of the mean NH$_3$ surface concentrations of the MAN network quite well, with low NH$_3$ surface concentrations near the coast, and increasing values towards the eastern of the Netherlands. The sum of the absolute differences between the warm season mean NH$_3$ surface concentrations in a cubic meter from MAN and LOTOS-EUROS was 444.7 µg in 2013 and 494.3 µg in 2014. The sum of the absolute differences between the warm season mean NH$_3$ surface concentrations from MAN and IASI was slightly higher in both years, amounting to 512.1 µg in 2013 and 513.6 µg in 2014.





**Figure 8: Comparison of the warm season (April-September) mean NH$_3$ surface concentration in 2013 (a, b, c, d) and in 2014 (e, f, g, h) from LOTOS-EUROS and derived using IASI. The corresponding warm season mean NH$_3$ surface concentrations measured by the MAN stations are plotted on top of the left figures. The right figures depict the differences between the two.**





Figure 9 shows a scatterplot of all the MAN monthly and warm season mean $NH_3$ surface concentrations versus the corresponding LOTOS-EUROS and IASI-derived concentrations in 2013 and 2014. The LOTOS-EUROS and the MAN network monthly mean $NH_3$ surface concentration show a moderate positive linear relationship (r =0.5 in 2013, r = 0.46 in 2014). The correlation of the IASI-derived and the MAN surface concentrations is somewhat

weaker in both years (r = 0.40 in 2013, r = 0.38 in 2014). The warm season mean $NH_3$ surface concentrations derived from IASI show a similar to slightly stronger correlation with the MAN observation (r = 0.59 in 2013, r = 0.54 in 2014) compared to the the warm season mean $NH_3$ surface concentrations from LOTOS EUROS (r = 0.54 in 2013, r = 0.54 in 2014).

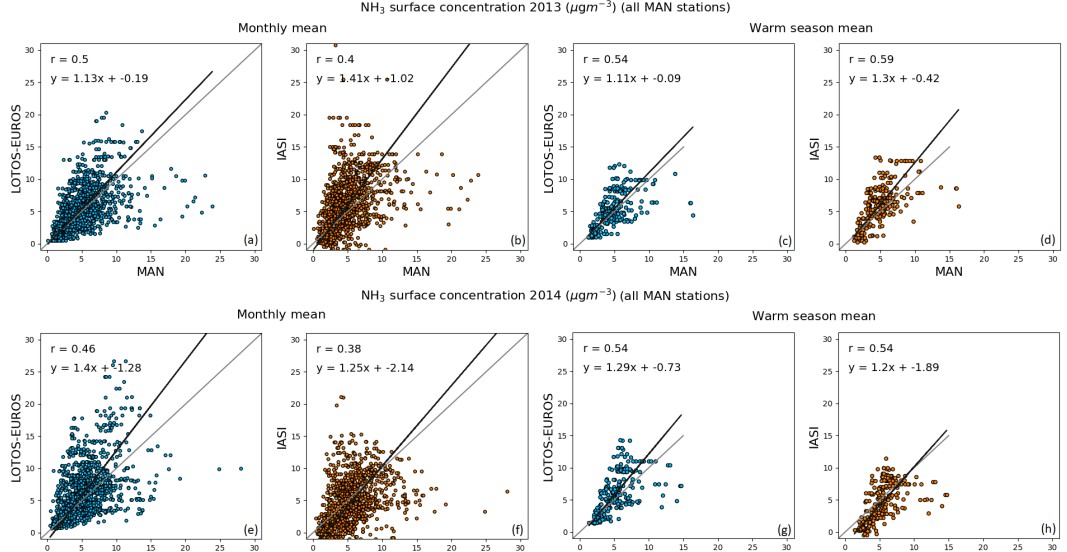

**Figure 9: Comparison of the monthly mean (left) and warm season (April-September) mean (right) $NH_3$ surface concentrations measured by the MAN stations and the corresponding $NH_3$ surface concentrations from LOTOS-EUROS (blue dots) and inferred from IASI (orange dots) in 2013 (top) and 2014 (bottom).**

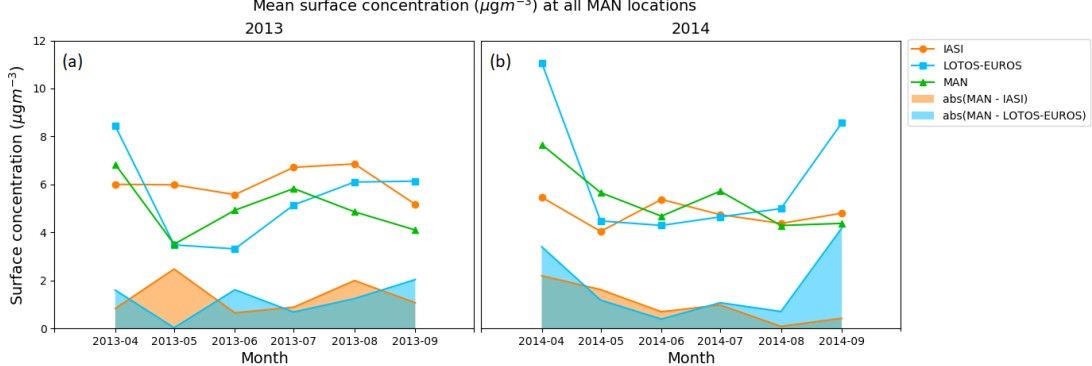

**Figure 10: Mean of the $NH_3$ surface concentrations at all MAN locations per month (green line), and the coinciding $NH_3$**

**surface concentrations from LOTOS-EUROS (blue line) and IASI (orange line) in 2013 (a) and 2014 (b). The absolute**



**differences between MAN and LOTOS-EUROS are shown in blue and the absolute differences between MAN and IASI are shown in orange.**

Figure 10 shows the mean NH$_3$ surface concentration of all MAN stations per month, and the corresponding mean NH$_3$ surface concentration from LOTOS-EUROS and IASI at the same locations. The absolute differences per

month are plotted in the same figure in blue (LOTOS-EUROS vs MAN) and orange (IASI-derived vs MAN). The time profile of the MAN stations monthly mean peaks in April in both years. In 2013 the mean NH$_3$ surface concentration of all MAN stations increases from May and peaks in July and then decreases towards the ending of the year. In 2014 there is also another peak in July, followed by a decrease. The LOTOS-EUROS mean NH$_3$ surface concentration at the same locations are higher than the ones measured by the MAN stations in April, August and

September in both years. The mean NH$_3$ surface concentration derived from IASI is lower than the ones from LOTOS-EUROS and MAN in April, and peaks in August in 2013 and in June in 2014. The sum of the absolute differences of the mean NH$_3$ surface concentrations in a cubic meter at all MAN locations between LOTOS-EUROS and MAN amounts to 7.2 µg in 2013 and 10.9 µg in 2014. The sum of the absolute differences between IASI and MAN was somewhat larger in 2013, amounting to 7.9 µg, but considerably smaller in 2014, amounting to 6.0 µg.

| | month | LOTOS-EUROS | | | | IASI-derived | | | |
|---|---|---|---|---|---|---|---|---|---|
| | | r | slope | intercept | RMSD | r | slope | intercept | RMSD |
| MAN | 04-2013 | 0.53 ↑ | 1.48 | -1.41 | 4.33 | 0.46 | 1.05 ↑ | -1.08 ↑ | 3.37 ↑ |
| | 05-2013 | 0.48 ↑ | 0.92 ↑ | 0.30 | 1.95 ↑ | 0.44 | 1.69 | 0.04 ↑ | 3.94 |
| | 06-2013 | 0.59 | 0.70 ↑ | -0.06 ↑ | 2.66 ↑ | 0.59 | 1.42 | -1.19 | 3.23 |
| | 07-2013 | 0.48 ↑ | 0.71 | 0.94 | 3.32 ↑ | 0.44 | 1.15 ↑ | -0.06 ↑ | 4.18 |
| | 08-2013 | 0.49 | 0.89 ↑ | 1.67 | 3.37 ↑ | 0.49 | 1.15 | 1.11 ↑ | 4.03 |
| | 09-2013 | 0.40 ↑ | 1.45 ↑ | 0.15 ↑ | 3.47 ↑ | 0.25 | 3.05 | -7.48 | 6.09 |
| | 04-2014 | 0.52 ↑ | 1.75 | -2.80 | 5.66 | 0.35 | 0.98 ↑ | -2.03 ↑ | 4.24 ↑ |
| | 05-2014 | 0.39 | 0.80 | -0.10 ↑ | 2.78 ↑ | 0.46 ↑ | 1.08 ↑ | -2.12 | 3.17 |
| | 06-2014 | 0.70 | 0.87 ↑ | 0.12 ↑ | 2.08 ↑ | 0.71 ↑ | 1.41 | -1.44 | 2.74 |
| | 07-2014 | 0.56 | 0.76 | 0.18 ↑ | 2.74 ↑ | 0.56 | 1.08 ↑ | -1.79 | 3.13 |
| | 08-2014 | 0.47 | 1.31 ↑ | -0.57 ↑ | 2.44 ↑ | 0.47 | 1.50 | -2.09 | 2.58 |
| | 09-2014 | 0.28 ↑ | 1.22 ↑ | 3.42 ↑ | 6.03 ↑ | 0.12 | 1.87 | -3.73 | 6.23 |

**Table 4: Month by month comparison of the correlation coefficient (r), slope and intercept of the monthly mean NH$_3$ surface concentrations of the MAN stations (x-axis) and the coinciding monthly mean LOTOS-EUROS and IASI-derived NH$_3$ surface concentrations (y-axis). The green arrows denote which of the two (LOTOS-EUROS or IASI) gives the most desirable values. The green arrows are attributed to either LOTOS-EUROS or IASI based on the following criteria: highest r, slope closest to 1, intercept closest to 0 and smallest RMSD.**

Table 4 shows the correlation coefficient r, the slope and the intercept of the MAN surface concentrations versus the LOTOS-EUROS and the IASI-derived NH$_3$ surface concentrations for the warm months in 2013 and 2014. In 2013, the IASI-derived NH$_3$ surface concentrations do not show a clear improvement compared to the LOTOS-EUROS



NH$_3$ surface concentrations in any of the months. In 2014, the IASI-derived NH$_3$ surface concentrations compared slightly better to the MAN observations in May and June.

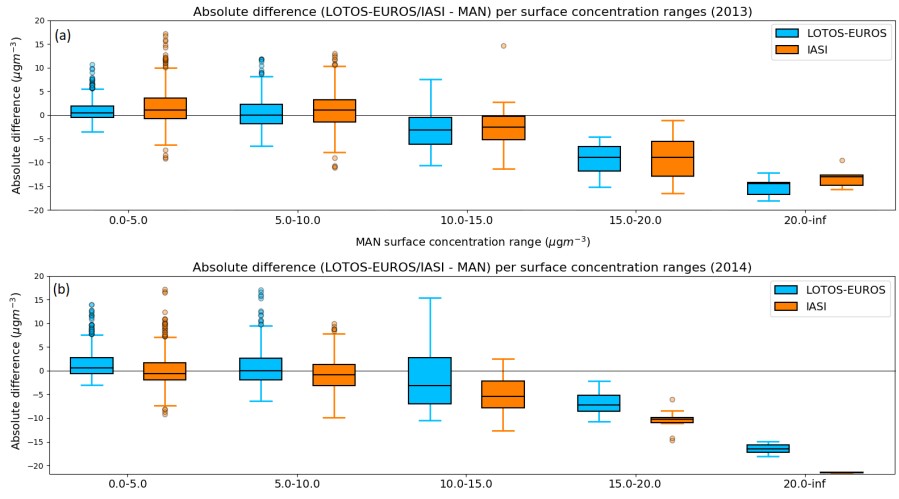

**Figure 11: The absolute differences between the monthly mean NH$_3$ surface concentrations modelled in LOTOS-EUROS**

5 **(blue) and derived from IASI (orange) and the monthly mean NH$_3$ surface concentrations measured by the MAN stations in the warm season (April-September) in 2013 (a) and 2014 (b), grouped as function of the MAN monthly mean NH$_3$ surface concentrations. The black line indicates the median, the edges of the boxes indicate the 25$^{th}$ and the 75$^{th}$ percentiles (Q1 and Q2), the whiskers indicate the full range of the absolute differences (Q1 – 1.5\*IQR and Q3 + 1.5\*IQR) and the dots indicate the outliers values that lie outside the whiskers.**

In order to test the performance of the LOTOS-EUROS and IASI-derived NH$_3$ surface concentrations as a function of concentration level, the data is grouped based on different MAN NH$_3$ surface concentration ranges. Figure 11 shows the grouped absolute differences between the monthly mean NH$_3$ surface concentrations measured by MAN and the monthly mean NH$_3$ surface concentrations from LOTOS-EUROS and derived from IASI. For the low MAN

15 concentration ranges (0-10 μgm$^{-3}$) the corresponding LOTOS-EUROS monthly mean NH$_3$ surface concentrations seem to agree fairly well in both years. For higher MAN concentration ranges (>10 μgm$^{-3}$) the LOTOS-EUROS model seems to underestimate the monthly mean NH$_3$ surface concentrations. In 2013, the IASI-derived NH$_3$ surface concentrations were relatively higher than the ones from LOTOS-EUROS for all concentration levels. The opposite is true in 2014, where the IASI-derived NH$_3$ surface concentrations were relatively lower than the ones from

20 LOTOS-EUROS. The differences between the LOTOS-EUROS and the IASI-derived NH$_3$ surface concentrations in the Netherlands can thus not be assigned to specific concentration levels.



In summary, the comparison with the MAN stations does also not show any significant, or consistent improvement in the IASI-derived $NH_3$ surface concentrations compared to the originally modelled $NH_3$ surface concentrations from LOTOS-EUROS.

### 4.1.3 Summary of the comparison with in-situ measurements

The comparison of the LOTOS-EUROS and IASI-derived $NH_3$ surface concentrations with the European EMEP network showed that the IASI-derived $NH_3$ surface concentrations slightly improved on a monthly basis compared to the originally modelled concentrations from LOTOS-EUROS. Moreover, the improvement became more pronounced when comparing on a seasonal basis (mean April-September). For the Netherlands, however, both the comparison with the LML and the MAN network did not show any significant, or consistent improvement in the IASI-derived $NH_3$ surface concentrations compared to the originally modelled $NH_3$ surface concentrations from LOTOS-EUROS.

The differences between Europe and the Netherlands could be explained by the fact that most of the European scale stations are located in background regions, with relatively well-mixed and low $NH_3$ concentrations, whereas most stations in the Netherlands are located in, or nearby, regions with relatively higher $NH_3$ concentrations. As a result, the vertical profile shapes in LOTOS-EUROS in the Netherlands are more complex and variable in time, as this region is influenced by a constantly changing combination of transport, emission and deposition. The use of an inadequate vertical profile to derive $NH_3$ surface concentrations from IASI could lead to an erroneous redistribution of the total amount of measured $NH_3$, therewith worsening the comparability with in-situ measurements. On the contrary, the vertical profile shapes in background regions are more stable and constant in time, and therefore more likely to be adequately described by the LOTOS-EUROS model.

#### 4.1.3.1 Side-note on validation with in-situ measurements

The differences between the in-situ measurement and the LOTOS-EUROS model and IASI can partially be explained by their discrepancy in terms of spatial representation, which limits their comparability to some extent. The footprint of the in-situ measurements is relatively small and easily influences by local factors, whereas the model and the satellite provide us with a mean value over a much larger area. The two high-concentration stations of the LML network in the Netherlands, Vredepeel and Wekerom, are for instance influenced by nearby emission sources which cannot be resolved by regional models at the current resolution.

### 4.2 $NH_3$ dry deposition flux

#### 4.2.1 Europe

The monthly mean dry $NH_3$ deposition flux has been computed for the warm months (April to September) of the year 2013 and 2014. Figure 12 shows the warm season mean dry $NH_3$ deposition flux (kg N ha$^{-1}$yr$^{-1}$) originally modelled in LOTOS-EUROS and the flux inferred from IASI combined with the LOTOS-EUROS model (which will be called the 'IASI-derived' flux from now on). The modelled warm season mean dry $NH_3$ deposition fluxes



from LOTOS-EUROS were very similar in 2013 and 2014. Figure 13 shows the absolute and relative differences between the two fluxes. In 2013, the IASI-derived fluxes in the Netherlands and Belgium were higher than the modelled fluxes, depicting that here the IASI-observed $NH_3$ total columns were thus higher than the modelled total columns in LOTOS-EUROS. In other areas such as Germany, and large parts of Central Europe, mainly in Poland,

Belarus and Romania the IASI-derived fluxes were higher than the modelled fluxes. In 2014, the IASI-derived fluxes were much higher than the modelled flux in some parts of Central Europe, mainly in Poland and the Czech Republic, and in some parts of the United Kingdom , for instance North-Ireland. The IASI-derived fluxes were much lower than modelled fluxes in Switzerland, the Po Valley in Italy and the northern part of Turkey, both in 2013 and 2014. At these locations the IASI-observed $NH_3$ total columns were thus consistently lower than the

modelled total columns in LOTOS-EUROS. An explanation for the differences could be that the data for the emissions at these locations are inadequate or that for instance the atmospheric transport and/or stability of $NH_3$ in the model is not modelled correctly.

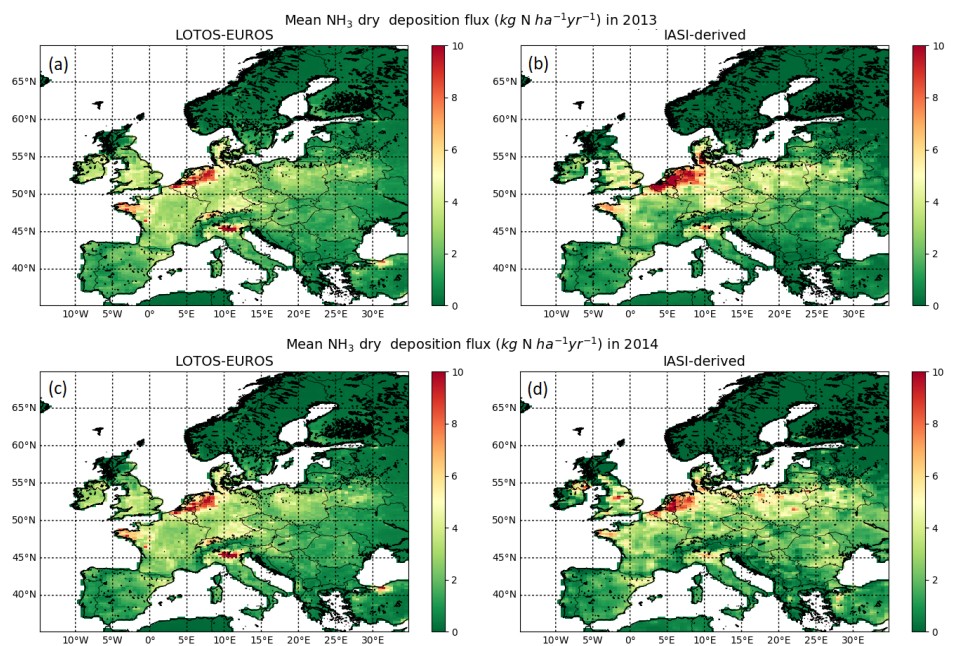

**Figure 12: The warm season (April to September) mean $NH_3$ dry deposition modelled in LOTOS-EUROS (left) and**

**inferred from IASI (right) in kg N ha$^{-1}$yr$^{-1}$ in 2013 (top) and 2014 (bottom).**





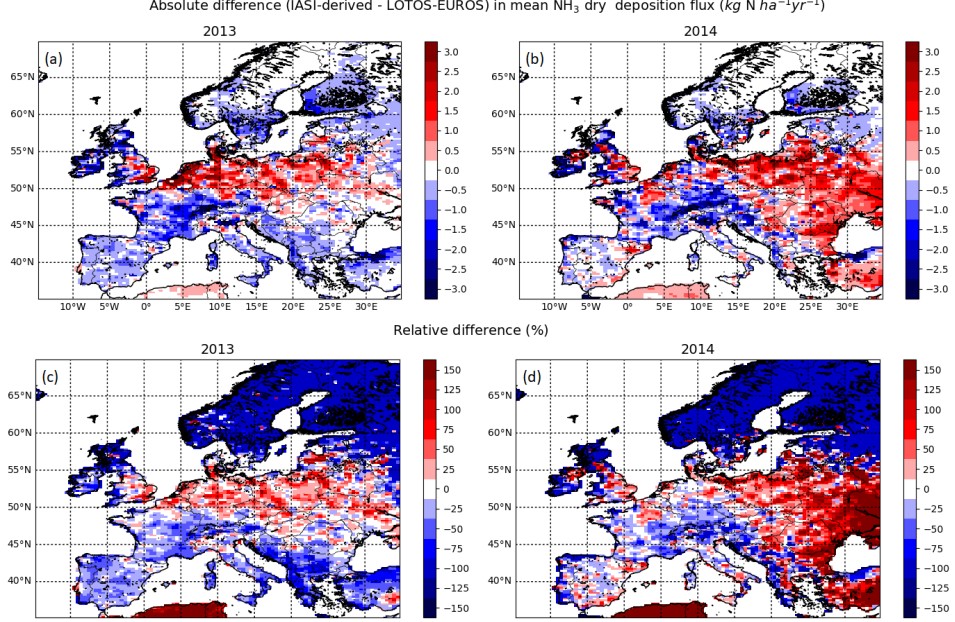

**Figure 13: The absolute (top) and relative (bottom) differences in warm season (April to September) mean NH$_3$ dry deposition modelled in LOTOS-EUROS and inferred from IASI in 2013 (left) and 2014 (right).**

### 4.2.2    The Netherlands

5    The warm season dry NH$_3$ deposition fluxes in the Netherlands modelled in LOTOS-EUROS and derived from IASI are shown in Figure 14. Figure 14 shows that the warm season mean NH$_3$ dry deposition flux in LOTOS-EUROS is fairly equal in 2013 and 2014, whereas the IASI-derived flux varies quite a lot. The IASI-derived flux is higher than the modelled flux in 2013, and lower than the modelled flux in 2014. The IASI-observed NH$_3$ total columns in the Netherlands were thus in general somewhat higher than the modelled NH$_3$ columns in 2013, and somewhat lower

10    than the modelled NH$_3$ columns in 2014.



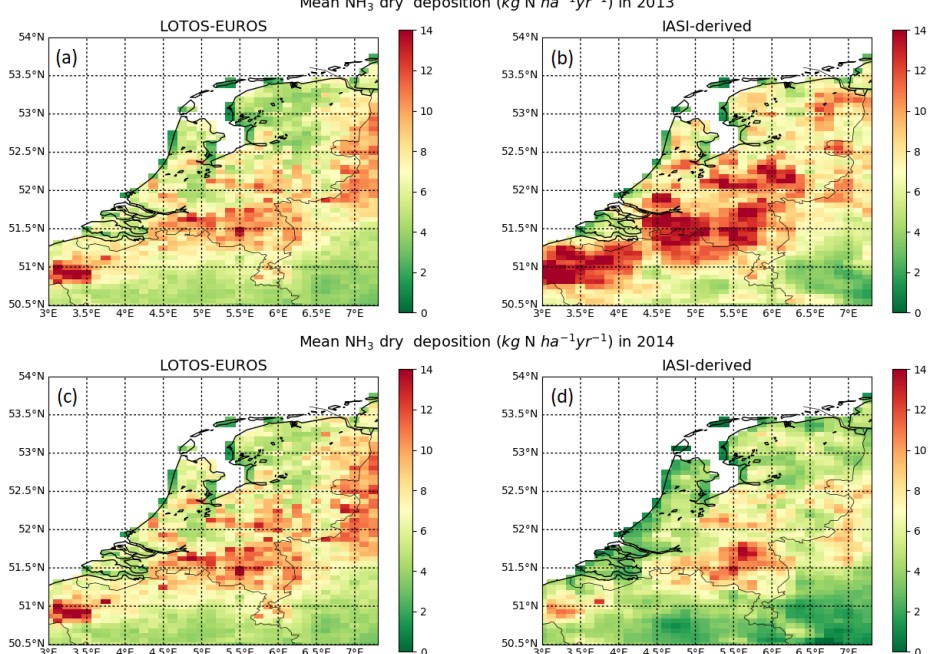

**Figure 14: The warm season (April to September) mean NH₃ dry deposition in the Netherlands modelled in LOTOS-EUROS (left) and inferred from IASI (right) in kg N ha⁻¹yr⁻¹ in 2013 (top) and 2014 (bottom).**





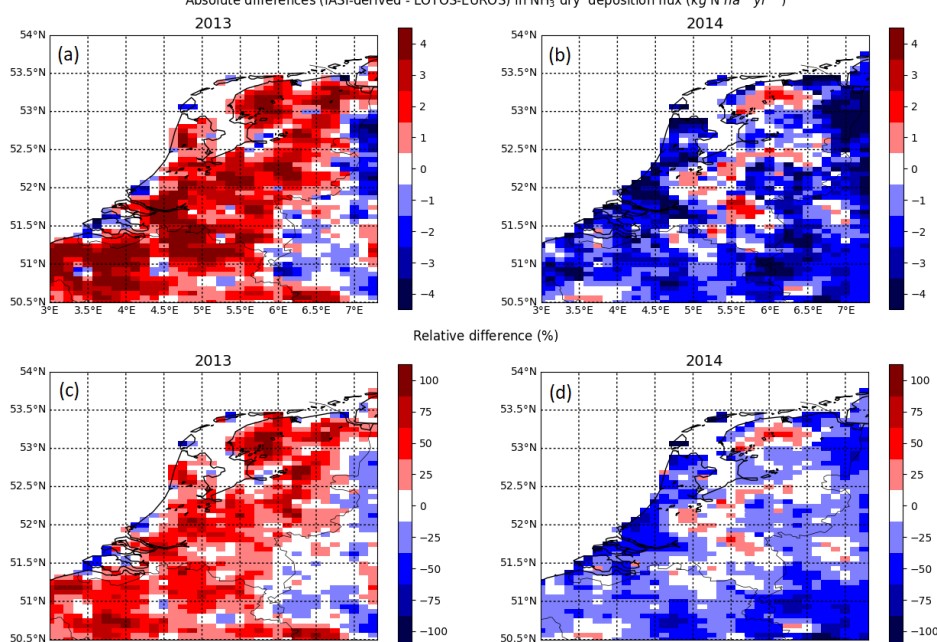

**Figure 15: The absolute (top) and relative (bottom) differences in warm season (April to September) mean NH$_3$ dry deposition in the Netherlands modelled in LOTOS-EUROS and inferred from IASI in 2013 (left) and 2014 (right).**

Figure 15 depicts the absolute and relative differences between the warm season mean dry NH$_3$ deposition fluxes modelled in LOTOS-EUROS and derived from IASI in 2013 and 2014. In 2013, the main differences between the two occur in the central and northernmost parts of the Netherlands, where the IASI-derived fluxes are clearly higher than the modelled ones. The largest part of the Netherlands the IASI-derived fluxes are higher than the LOTOS-EUROS fluxes. In 2014, the IASI-derived deposition fluxes are lower than the modelled fluxes for the largest part of the Netherlands, except for the center and the northernmost part of the Netherlands.

## 4.3. LOTOS-EUROS sensitivity study

The results of the sensitivity runs are summarized in Figure 16, Figure 17 and Figure 18. Figure 16 shows the relative changes in the warm season mean terrestrial dry NH$_3$ deposition flux over Europe modelled in LOTOS-EUROS (a) and derived from IASI (b) in 2014 for the different model runs. The mean LOTOS-EUROS dry NH$_3$ deposition over the land cells in the modelling grid in 2014 was 1.76 kg N ha$^{-1}$ yr$^{-1}$. The mean IASI-derived dry NH$_3$ deposition flux was somewhat higher, namely 2.20 kg N ha$^{-2}$ yr$^{-1}$.

The largest change in the modelled dry NH$_3$ deposition flux in LOTOS-EUROS was obtained by variations in the MACC-III NH$_3$ emissions and the smallest change was obtained by applying variations to the wet deposition scavenging coefficient G$_{scav}$. The changes in the dry deposition velocity V$_d$ led to the biggest changes in the final IASI-derived dry NH$_3$ deposition flux . The effect appears to be amplified compared to the effect on the LOTOS-



EUROS dry $NH_3$ deposition flux. The effect of the MACC-III $NH_3$ emissions appears to be damped, whereas the effect of the MACC-III $NO_X$ and $SO_2$ emissions is amplified. The signs of the changes in the dry $NH_3$ deposition flux derived by IASI have flipped as a result of the changes in MACC-III $NH_3$, MACC-III $NO_X$ and $SO_2$ and $G_{scav}$. The LOTOS-EUROS dry $NH_3$ deposition is one to one sensitive to emission changes in $NH_3$, whereas for IASI-derived dry $NH_3$ deposition this is much less. The IASI-derived dry $NH_3$ deposition in turn changes one to one with the $V_d$.

The variations in LOTOS-EUROS dry $NH_3$ deposition are a result of daily and monthly variations in emissions. The variations in the IASI-derived dry $NH_3$ deposition are also a result of these variation, but on top of this also include an effect of the overpass time.

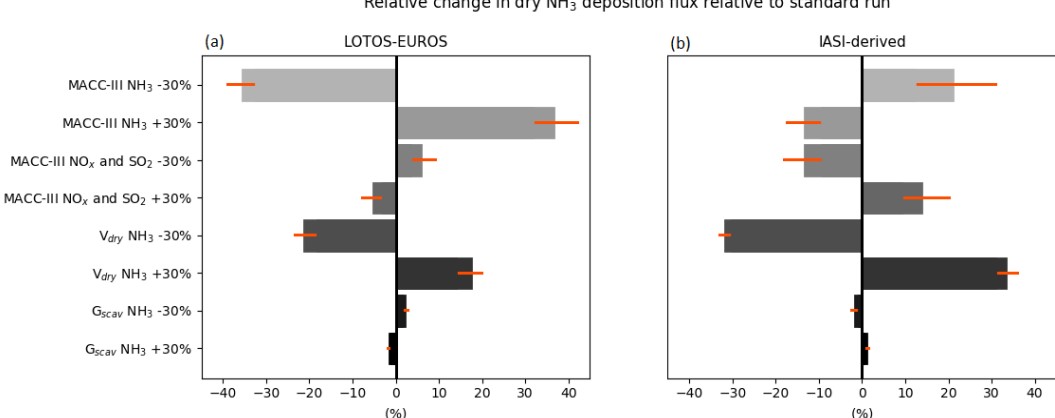

**Figure 16: The median change (%) in the terrestrial $NH_3$ dry deposition flux in 2014 in (kg N ha$^{-1}$ yr$^{-1}$) from LOTOS-EUROS (a) and IASI-derived (b), resulting from different perturbations of model inputs of LOTOS-EUROS. The orange lines indicate the 25$^{th}$ and the 75$^{th}$ quartiles.**



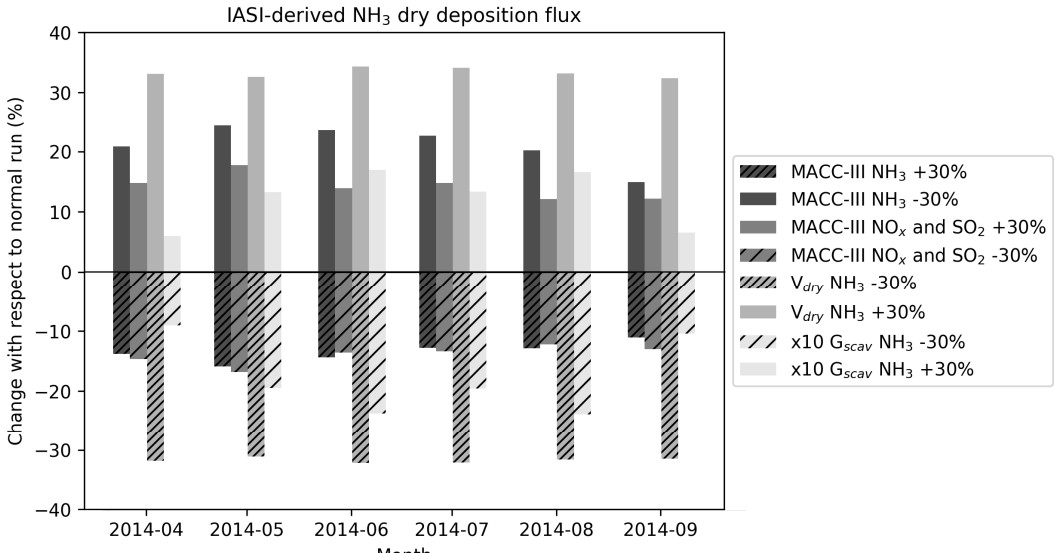

**Figure 17: The change (%) in the monthly mean IASI-derived NH₃ dry deposition flux resulting from different perturbations of the LOTOS-EUROS model.**

Figure 17 shows the changes (%) of monthly mean IASI-derived dry NH$_3$ deposition fluxes in 2014 resulting from the different LOTOS-EUROS sensitivity runs. Note that the effect of the runs with changes in wet deposition through variations of the gas scavenging coefficient for NH$_3$ are enlarged by a factor 10. We see that the changes with respect to the standard LOTOS-EUROS run are in general constant over the months. The least variation is observed for the runs with changed V$_{dry}$ values, that all resulted in a change of ~31% per month. The runs with adjusted MACC-III emissions of NH$_3$ and emissions of NO$_x$ and SO$_2$ led to largest changes in May and the smallest changes in September. The maximum difference between months is 9.5% and 5.6%, respectively, for the runs with adjusted NH$_3$ and the runs with adjusted NO$_x$ and SO$_2$ values. The runs with changed values of G$_{scav}$ for NH$_3$ seems to be affected most by changing weather conditions, which resulted in the relatively largest variation per month. However, because the changes in the IASI-derived dry NH$_3$ deposition fluxes are small (-2.4 to +1.7%), we now continue to look at yearly changes.





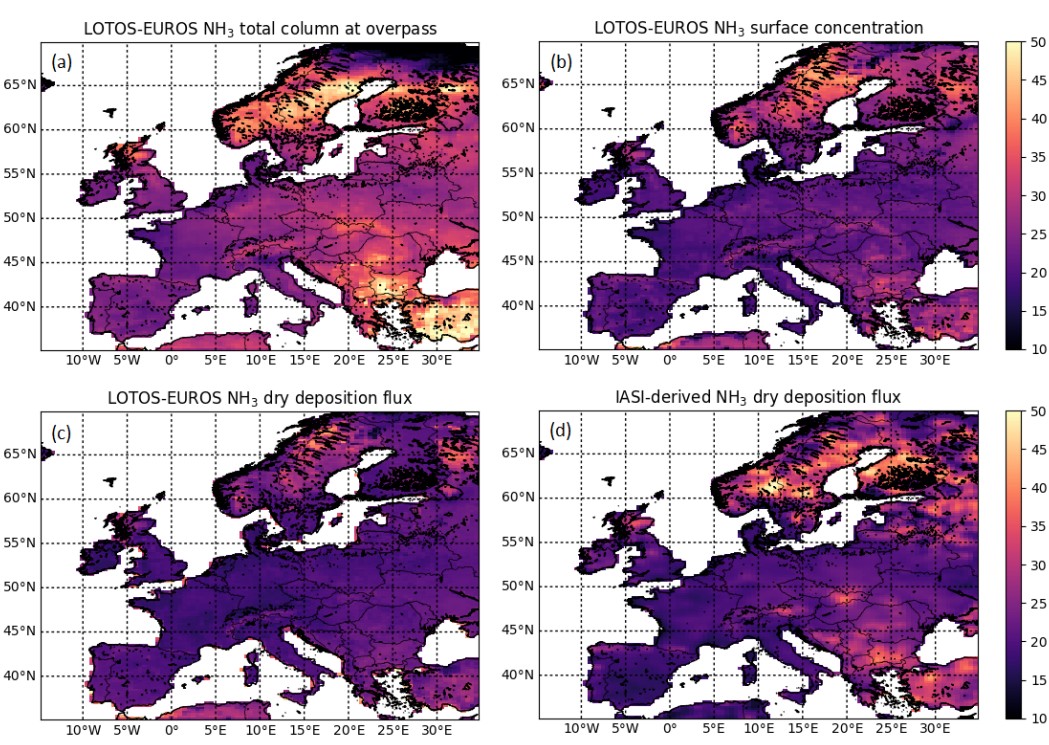

**Figure 18: The relative standard deviation (%) of the warm season mean output of all perturbed runs and the associated dry deposition estimate inferred from IASI in 2014. Figure (a) shows the LOTOS-EUROS NH$_3$ total column concentration at overpass, (b) the LOTOS-EUROS NH$_3$ surface concentration, (c) the NH$_3$ dry deposition flux in LOTOS-EUROS and (d) the resulting IASI-derived NH$_3$ dry deposition flux.**

Figure 18 shows the relative standard deviation (%) of all 8 sensitivity runs for Europe. The bottom-right figure (d) shows the relative standard deviation of the final IASI-derived dry NH$_3$ deposition flux. The relative standard deviation varies from ~20% to ~50% throughout Europe. The smallest variations can be seen in the south-western and central parts of Europe. The highest variations of ~40 - 50% are mainly found in long-distance transport areas with low NH$_3$ concentrations and deposition fluxes, such as Scandinavia, and in areas with high aerosol precursor emissions, such as the Balkans.

## 5. Discussion

In this paper we determined the applicability and the limitations of the method suggested by Nowlan et al. (2014) for the derivation of NH$_3$ surface concentrations and dry deposition fluxes across Europe. A comparison of the LOTOS-EUROS modelled and IASI-derived NH$_3$ surface concentrations with in-situ measurements of the EMEP network on a European scale and the LML and MAN network in the Netherlands has been made. Although there appeared to be some improvements in the IASI-derived NH$_3$ surface concentrations compared to the modelled LOTOS-EUROS

NH$_3$ surface concentrations, mainly in background regions, there did not seem to be any significant, consistent improvement. Also, the timing of the IASI-derived NH$_3$ surface concentrations did not show better correspondence with the in-situ observations than the modelled NH$_3$ surface concentrations. Consequently, as the dry NH$_3$ deposition fluxes are directly derived from the NH$_3$ surface concentrations, no significant improvement is expected

here either. On top of this, the sensitivity study using eight input parameters important for NH$_3$ dry deposition modelling showed that the effect of model uncertainties on the IASI-derived dry NH$_3$ deposition fluxes is amplified by the estimation procedure compared to the effect on the model simulations itself. The final IASI-derived dry NH$_3$ deposition fluxes can vary ~20% up to ~50% throughout Europe as a result of model uncertainties.

The method used to derive the NH$_3$ surface concentrations and dry deposition fluxes from IASI observations is

based on various assumptions. For one, the method assumes that the relationship between NH$_3$ concentration and the dry deposition fluxes is linear, whereas this relationship is in reality non-linear. In fact, these quantities can even be anti-correlated with highest surface concentrations during the night when the atmosphere is stable and the exchange is limited. The non-linearity is further enhanced by the compensation point of NH$_3$. For our purpose, focusing on a single time of the day using monthly data, however, approximating this concentration-flux relationship

by a linear curve may seem reasonable for concentration regimes below the saturation point. For higher NH$_3$ surface concentrations the current approach will likely lead to overestimated dry deposition fluxes. Moreover, this study includes the impact of the compensation point of NH$_3$ through the dry deposition scheme in LOTOS-EUROS. Although the uncertainties are relatively large as the compensation points derived is based on relatively few observations (e.g. (Wichink Kruit et al., 2007)) , we feel that the inclusion of the compensation point is a strong

point of this study.

Moreover, the approach by Nowlan also assumes that the NH$_3$ total column concentrations measured by IASI serve as a direct proxy of the NH$_3$ surface concentrations, whereas in reality, the relationship between the two is influenced by various different factors, including the vertical distribution of NH$_3$ and the satellites sensitivity. There are already quite some uncertainties involved with the vertical distribution of NH$_3$ and therefore tower measurement

campaigns (Dammers et al., 2017a;Li et al., 2017a) are very important to strengthen our understanding. Dammers et al. (2017a) for instance showed that the daytime boundary layer is well-mixed, which supports the choice for a model that uses the mixed layer approach such as LOTOS-EUROS. Li et al. (2017b) showed that there is a clear seasonal variation in the vertical distribution of NH$_3$ and that the slope of the NH$_3$ concentration gradient varies throughout the year. During winter Li et al. (2017b) observed relatively high NH$_3$ ground concentrations due to

potential trapping of NH$_3$ emissions in a shallow winter boundary layer, and reduced NH$_3$ concentrations higher up the column. In these types of situations the IASI-satellite instrument potentially misses high NH$_3$ ground concentrations because of the lack of sensitivity to the lower parts of the boundary layer. The computation of averaging kernels for IASI could help to indicate more precisely where the sensitivity lies and how the measured total columns are distributed. Moreover, further development and validation of the IASI retrieval may help to

improve our understanding of the satellites product, therewith also increasing its applicability.



The method also assumes that the timing and distribution of the emissions in the LOTOS-EUROS model closely represent reality, as the ratio between the retrieved and the modelled ammonia burden is used at overpass time. The accuracy of the seasonal variation in NH$_3$ emissions in LOTOS-EUROS is therefore of great importance. The reliability of yearly dry NH$_3$ deposition estimates using our method is limited by the lack of high-quality IASI

observations during the cold season. As a result, derivation of yearly IASI-derived NH$_3$ dry deposition estimates may differ substantially depending on whether or not the spring maximum peak occurs in the satellite-observed months (April – September). Skjøth et al. (2011) presented the seasonal variation and the distribution of NH$_3$ emissions for different European countries per agricultural source, and showed for instance that approximately half of the NH$_3$ emissions from spring fertilization is usually emitted in March. As the spring fertilization amounts to

~20-50% of the yearly total NH$_3$ emissions, this may result in a variation of the same magnitude on the subsequent deposition estimates. Improvement of the seasonal variation in NH$_3$ emissions in LOTOS-EUROS could be used to fill-in this gap and lead to a more accurate representation of reality. Skjøth et al. (2011) showed that the implementation of a dynamic NH$_3$ emission model for different agricultural sources may result in considerable model performance improvements when high-quality activity data and information on spatial distributions of

emissions is available. Furthermore, Hendriks et al. (2016) showed that the use of manure transport data for ammonia emission time profiles lead to additional model improvements and a better representation of the spring maximum.

Moreover, mismatches between the actual and modelled diurnal variations in NH$_3$ emission could also easily lead to large differences in the IASI-derived dry NH$_3$ deposition estimates. As an illustration, Sintermann et al. (2016) for

instance measured NH$_3$ emissions from an agricultural surface after slurry application and showed that ~80% of the total NH$_3$ was emitted within 2 hours. Combined with the short-lifetime of NH$_3$ there is a possibility that the IASI-instrument completely misses these kind of events if they occur after its overpass. A possible way to reduce the impact of the diurnal variation is to combine observations from IASI with observations from other satellites with different overpass times, for instance NH$_3$ observations from the CrIS satellite instrument (Shephard and Cady-

Pereira, 2015).

At this stage we can conclude that the IASI-derived NH$_3$ deposition fluxes do not show a strong improvements compared to modelled NH$_3$ deposition fluxes and there is future need for better, more robust, methods to derive NH$_3$ dry deposition fluxes. This could potentially be achieved by further integration of existing in-situ- and satellite data into models with special attention to data representativeness, for instance by means of data-assimilation. In addition,

there is a need for a better understanding of the surface exchange of NH$_3$ for different land use types. Model parameterizations of the surface exchange of NH$_3$ are at the moment based on a limited number of direct flux measurements, and more measurements could definitely improve this. Also, better understanding of the timing and distribution of NH$_3$ emissions could lead to considerate improvements in modelled emissions fields and consequently deposition fields from CTMs.





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
