# Peer review of "Technical note: How are NH3 dry deposition estimates affected by combining the LOTOS-EUROS model with IASI-NH3 satellite observations?"

_Atmospheric Chemistry and Physics, 2018_

## Referee Comment (RC1) · Anonymous Referee #2 · 11 Jul 2018

The paper investigates the effects of using satellite-derived NH3 levels in a chemistry transport model on the modeled NH3 concentrations and deposition fluxes. The paper is interesting and easy to follow. I am in favor of its publication in ACP provided it address the points below.

General Comments

- Can the authors elaborate on why results are different in the two years? - Is meteorology playing a role here? Is it possible to validate the meteorology to enrich the discussions? - In addition, both the original and IASI inferred NH3 concentrations are overestimated both years. Can the authors discus why? Is it overestimation in emissions or underestimation in deposition? - Why are the deposition fluxes not evaluated against observations?

Technical comments

Page 1 Line 33: . . .do not show strong improvements. . ..

Page 2, Line 30: . . .ALLOW us to . . .. . .

Section 2.2. needs some more explanation of how the uncertainty is calculated.

Section 2.4.1. needs more information on the temporal variation of emissions, in particular $NH_3$.

Page 8, Line 16: Erisman (1993) estimated. . ..

Page 9, Line 16: . . .dry deposition fluxes IN Eq. (3):

---

## Referee Comment (RC2) · Anonymous Referee #1 · 16 Jul 2018

This study deals with dry deposition of NH3 using the deposition scheme currently implemented in Lotos-EUROS model as well as the remote sensing retrieval from IASI. it is overall a neat work, although a bit limited in the applicability and range of conclusions. I suggest the editor to grant publication of this work as technical contribution to ACP, conditioned to some minor improvements:

- my main comment is related to the derivation of IASI concentration and fluxes. it seems to me that these quantities rely heavily on the modelled outcome. This is fine of course, but I wonder about the robustness of results such as: 'There appears to be some minor improvements inthe IASI-derived NH3 surface concentrations compared to

the modelled NH3 surface concentrations from LOTOS-EUROS on a monthly basis...'. I am might missing something here - or just haven't understood fully your approach - but from the paper it'd seem that you are comparing two highly dependent variables. if that is the case then the conclusion that the two sets of results are quite similar is kind of given; otherwise please consider restructuring the description of the methodology to leave no doubts.

- the examined periods (two warm seasons) might be a but limited to screen out meteorology effects. and/or episodic event. Please comment on this

- please consider 'Modeled deposition of nitrogen and sulfur in Europe estimated by 14 air quality model systems: evaluation, effects of changes in emissions and implications for habitat protection' by Vivanco et al, 2018 (ACPD), which also includes deposition results from LOTOS-EUROS.

- please consider a careful reading and editing of the entire manuscript. Although overall comprehensible, some sentences are a bit obscure and/or too long and/or redundant/unnecessary. For instance in the abstract: 'The aim of this paper is to determine for the applicability and the limitations of this method for NH3 using space-born observations of the Infrared Atmospheric Sounding Interferometer (IASI) and the LOTOS-EUROS atmospheric transport model.' Why not: 'The aim of this study is to determine the potential benefit of such a methodology to estimate the NH3 budget. Space-born observations from the Infrared Atmospheric Sounding Interferometer (IASI) and the LOTOS-EUROS atmospheric transport model are used.', or something on that line.

---

## Author Comment (AC1) · 27 Aug 2018

General authors comments The authors want to thank the reviewers for their encouraging and helpful comments. We have carefully addressed their reviews and revised the manuscript accordingly. We responded to each individual referee comment (marked as RC#) with an author comment (AC#). A pdf file of the author's comments and the manuscript changes are added as a supplement.

Manuscript changes (see supplement) - We have implemented the comments from the referees in the revised version of the manuscript (see the responses to Referee #1 & #2). - Lines with newly added content to the article are marked blue. - Lines that

are changed to improve the readability/conciseness of the article are marked green. - In addition to the above mentioned changes, we added a short acknowledgement to thank the applicable institutions for providing access to their datasets.

Anonymous Referee #1 RC1.1: "This study deals with dry deposition of NH3 using the deposition scheme currently implemented in Lotos-EUROS model as well as the remote sensing retrieval from IASI. it is overall a neat work, although a bit limited in the applicability and range of conclusions. I suggest the editor to grant publication of this work as technical contribution to ACP, conditioned to some minor improvements:"

AC1.1: The authors greatly thank the reviewer for the interest in our manuscript and encouraging comments.

RC1.2: "1- my main comment is related to the derivation of IASI concentration and fluxes. it seems to me that these quantities rely heavily on the modelled outcome. This is fine of course, but I wonder about the robustness of results such as: 'There appears to be some minor improvements in the IASI-derived NH3 surface concentrations compared to the modelled NH3 surface concentrations from LOTOS-EUROS on a monthly basis...'. I am might missing something here - or just haven't understood fully your approach - but from the paper it'd seem that you are comparing two highly dependent variables. if that is the case then the conclusion that the two sets of results are quite similar is kind of given; otherwise please consider restructuring the description of the methodology to leave no doubts."

AC1.2: It is indeed true that the two fields rely on the same model outputs and are therefore highly dependent. The NH3 surface concentrations that are adjusted by IASI observations overall lead to small changes over larger areas. This means that for the spatial variation of concentrations the LOTOS-EUROS model represents the spatial distribution rather well. However, in some occasions and on grid by grid basis we see considerable deviation. There are limited number of surface concentration observations and we do not see these differences in the comparison with the measurements.

Since we use the measurements as a reference and the comparison between model and observations changes very little, we conclude that the IASI overall does not lead to major improvements. The authors have rephrased parts of the method and results and added an additional statement in section '4.1.3 Summary of the comparison with in-situ measurements' to put extra emphasis on this:

"Generally, the modelled and the observed NH3 total columns match quite well. This means that the LOTOS-EUROS model represents the spatial distribution of NH3 rather well. There are some areas with large discrepancies between the two where we see considerable deviations in the modelled and the IASI-derived concentrations. Most of these areas, however, cannot be validated against measurements, because of the lack of measurements here. The changes in the comparison of the available measurements with modelled versus IASI-derived concentrations are therefore relativity small. Based on the measurements we have, we conclude that we do not see any significant improvement in the IASI-derived concentrations compared to the modelled concentrations."

RC1.3: " the examined periods (two warm seasons) might be a but limited to screen out meteorology effects. and/or episodic event. Please comment on this"

AC1.3: The authors agree that this is the case. We have therefore added a remark about this together with a short, general description of the meteorological circumstances in 2013 and 2014 at the end of section 4.2.2. See also our answer to RC2.2.

RC1.4: " please consider 'Modeled deposition of nitrogen and sulfur in Europe estimated by 14 air quality model systems: evaluation, effects of changes in emissions and implications for habitat protection' by Vivanco et al, 2018 (ACPD), which also includes deposition results from LOTOS-EUROS."

AC1.4: Thank you for the alerting us of the Vivanco et al, 2018 paper. We've added a reference in section 2.4.

RC1.5: " please consider a careful reading and editing of the entire manuscript. Although overall comprehensible, some sentences are a bit obscure and/or too long and/or redundant/unnecessary. For instance in the abstract: 'The aim of this paper is to determine for the applicability and the limitations of this method for NH3 using space-born observations of the Infrared Atmospheric Sounding Interferometer (IASI) and the LOTOS-EUROS atmospheric transport model.' Why not: 'The aim of this study is to determine the potential benefit of such a methodology to estimate the NH3 budget. Space-born observations from the Infrared Atmospheric Sounding Interferometer (IASI) and the LOTOS-EUROS atmospheric transport model are used.', or something on that line."

AC1.5: As suggested by the reviewer, the authors carefully re-read and edited the text in the entire manuscript. An English translator proofread and helped with editing the manuscript. The sentences that were changed are marked green in 'Manuscript changes'.

The authors appreciate the helpful comments of the reviewer.

Anonymous Referee #2 RC2.1: "The paper investigates the effects of using satellite-derived NH3 levels in a chemistry transport model on the modeled NH3 concentrations and deposition fluxes. The paper is interesting and easy to follow. I am in favor of its publication in ACP provided it address the points below."

AC2.1: The authors greatly appreciate the helpful and encouraging comments of the Referee.

RC2.2: "General Comments - Can the authors elaborate on why results are different in the two years? - Is meteorology playing a role here? Is it possible to validate the meteorology to enrich the discussions?"

AC 2.2: The authors do believe that meteorology is playing an important role in the inter-annual differences in our results, as it influences both the satellite retrieval and (to

a lesser extent) the model results. We looked at the meteorological circumstances in 2013-2014 have added the following section at the end of section 4.2.3.:

"The inter-annual variations of the modelled and IASI-derived flux differences (see Figure 13 and 15) could be related to different meteorological conditions. The annual global climate reports from the NOAA (National Oceanic and Atmosphere Administration) show that the mean temperatures in Europe were higher in 2014 than in 2013, especially in western Europe. This might have had an effect on the emissions, which is only limited taken into account by the model. The annual precipitation in both years was near average for Europe as a whole. However, if we zoom in to a more regional scale, we see that it was much wetter than average during the warm season in nearly all parts of the Balkan Peninsula and Turkey (NOAA, 2014, 2015). Figure 13 shows that the largest inter-annual variations on a European scale occur around the Black Sea: in Ukraine, but also in the eastern parts of the Balkan Peninsula and Turkey. Some of these regions thus coincide with regions that experienced heavy rainfall in 2014 and might have affected emission and deposition processes which are not taken into account by the model. This suggests that meteorological effects might indeed influence our results. However, the examined period of two warm seasons only is too short to draw a conclusion."

RC2.3:"- In addition, both the original and IASI inferred NH3 concentrations are overestimated both years. Can the authors discus why? Is it overestimation in emissions or underestimation in deposition?"

AC2.3: The modelled and the IASI-derived NH3 concentrations are indeed overestimated in emission areas. We added the following section to '4.1.3. Summary of the comparison with in-site measurements' to discuss this:

"In general, both the modelled and the IASI-derived concentrations seem to be overestimated in emission areas. This could potentially be related to the overpass time of the satellite. In high emission areas, the NH3 concentrations are more variable in time,

and the IASI observations might have an uncertain representativeness. Moreover, the measurements in high emission areas are generally more uncertain with regard to their spatial representativeness. Overall, these measurements can be more affected by local rather than regional sources. Generally, the modelled and the observed NH3 total columns match quite well. This means that the LOTOS-EUROS model represents the spatial distribution of NH3 rather well. There are some areas with large discrepancies between the two where we see considerable deviations in the modelled and the IASI-derived concentrations. Most of these areas, however, cannot be validated against measurements, because of the lack of measurements here. The changes in the comparison of the available measurements with modelled versus IASI-derived concentrations are therefore relativity small. Based on the measurements we have, we conclude that we do not see any significant improvement in the IASI-derived concentrations compared to the modelled concentrations."

From our results, it is impossible to tell whether the differences between the two fields are related to a systematic or significant deviation in either the emissions or the deposition. There are so many different uncertain variables involved in both the model and the measurements that it is impossible to pinpoint the most important reason. This would be a very interesting, and challenging topic for potential follow-up studies.

RC2.4: "- Why are the deposition fluxes not evaluated against observations?"

AC2.4: The authors would very much like to evaluate the model against observations of dry deposition fluxes against observations. However, the available NH3 dry deposition measurements in 2013 and 2014 are too limited to do a sensible model evaluation. There is certainly a need for more dry deposition measurements.

RC2.5: "Technical comments Page 1 Line 33: : : :do not show strong improvements: : :. Page 2, Line 30: : : :ALLOW us to : : :: : : Section 2.2. needs some more explanation of how the uncertainty is calculated. Section 2.4.1. needs more information on the temporal variation of emissions, in particular NH3. Page 8, Line 16: Erisman (1993)

estimated: : :. Page 9, Line 16: : : :dry deposition fluxes IN Eq. (3):"

AC2.5: Thank you for the technical comments. We added some additional explanation about the IASI uncertainty to section 2.2:

"The uncertainty estimate for each retrieved NH3 total column is an error propagation of the individual parameter uncertainties. Whitburn et al. (2016) showed in an error characterization that individual retrieved NH3 columns hold the smallest errors (∼25%) in the situation of a high NH3 concentration combined with a high thermal contrast. The error increases progressively when either of these lowers. In the case of a low NH3 concentration and a low thermal contrast, the errors can be as high as ∼270%."

We also added a short section about the temporal variations of the emissions in LOTOS-EUROS to section 2.4.1:

"LOTOS-EUROS uses a set of temporal factors (monthly, daily and hourly) to break down annual total emissions into hourly emissions. The time profile of a particular pollutant is an aggregation of the time-dependent emission strengths from different SNAP (Selected Nomenclature for Sources of Air Pollution) sources. The monthly NH3 emissions peak in March and then decrease, followed by another smaller peak in September. The daily NH3 emission strengths are re-distributed more or less evenly over the week. The hourly NH3 emission peak is reached at 13.00 h (Denier van der Gon et al., 2011)."

The authors sincerely thank the reviewer for his interest in our manuscript.

Please also note the supplement to this comment:
https://www.atmos-chem-phys-discuss.net/acp-2018-133/acp-2018-133-AC1-supplement.pdf

**Supplement:**

**General authors comments**

The authors want to thank the reviewers for their encouraging and helpful comments. We have carefully addressed their reviews and revised the manuscript accordingly. We responded to each individual referee comment (marked as RC#) with an author comment (AC#).

**Manuscript changes**

- We have implemented the comments from the referees in the revised version of the manuscript (see the responses to Referee #1 & #2).
- Lines with newly added content to the article are marked blue.
- Lines that are changed to improve the readability/conciseness of the article are marked green.
- In addition to the above mentioned changes, we added a short acknowledgement to thank the applicable institutions for providing access to their datasets.

**Anonymous Referee #1**

RC1.1: "This study deals with dry deposition of NH3 using the deposition scheme currently implemented in Lotos-EUROS model as well as the remote sensing retrieval from IASI. it is overall a neat work, although a bit limited in the applicability and range of conclusions. I suggest the editor to grant publication of this work as technical contribution to ACP, conditioned to some minor improvements:"
AC1.1: The authors greatly thank the reviewer for the interest in our manuscript and encouraging comments.

RC1.2: ''1- my main comment is related to the derivation of IASI concentration and fluxes. it seems to me that these quantities rely heavily on the modelled outcome. This is fine of course, but I wonder about the robustness of results such as: 'There appears to be some minor improvements in the IASI-derived NH3 surface concentrations compared to the modelled NH3 surface concentrations from LOTOS-EUROS on a monthly basis...'. I am might missing something here - or just haven't understood fully your approach - but from the paper it'd seem that you are comparing two highly dependent variables. if that is the case then the conclusion that the two sets of results are quite similar is kind of given; otherwise please consider restructuring the description of the methodology to leave no doubts.''
AC1.2: It is indeed true that the two fields rely on the same model outputs and are therefore highly dependent. The $NH_3$ surface concentrations that are adjusted by IASI observations overall lead to small changes over larger areas. This means that for the spatial variation of concentrations the LOTOS-EUROS model represents the spatial distribution rather well. However, in some occasions and on grid by grid basis we see considerable deviation. There are limited number of surface concentration observations and we do not see these differences in the comparison with the measurements. Since we use the measurements as a reference and the comparison between model and observations changes very little, we conclude that the IASI overall does not lead to major improvements. The authors have rephrased parts of the method and results and added an additional statement in section '4.1.3 Summary of the comparison with in-situ measurements' to put extra emphasis on this:

*"Generally, the modelled and the observed NH3 total columns match quite well. This means that the LOTOS-EUROS model represents the spatial distribution of NH3 rather well. There are some areas with large discrepancies between the two where we see considerable deviations in the modelled and the IASI-derived concentrations. Most of these areas, however, cannot be validated against measurements, because of the lack of measurements here. The changes in the comparison of the available measurements with modelled versus IASI-derived concentrations are therefore relativity small. Based on the measurements we have, we conclude that we do not see any significant improvement in the IASI-derived concentrations compared to the modelled concentrations."*

RC1.3: " the examined periods (two warm seasons) might be a but limited to screen out meteorology effects. and/or episodic event. Please comment on this"
AC1.3: The authors agree that this is the case. We have therefore added a remark about this together with a short, general description of the meteorological circumstances in 2013 and 2014 at the end of section 4.2.2. See also our answer to RC2.2.

RC1.4: " please consider 'Modeled deposition of nitrogen and sulfur in Europe estimated by 14 air quality model systems: evaluation, effects of changes in emissions and implications for habitat protection' by

Vivanco et al, 2018 (ACPD), which also includes deposition results from LOTOS-EUROS."

AC1.4: Thank you for the alerting us of the Vivanco et al, 2018 paper. We've added a reference in section 2.4.

RC1.5: " please consider a careful reading and editing of the entire manuscript. Although overall comprehensible, some sentences are a bit obscure and/or too long and/or redundant/unnecessary. For instance in the abstract: 'The aim of this paper is to determine for the applicability and the limitations of this method for NH3 using space-born observations of the Infrared Atmospheric Sounding Interferometer (IASI) and the LOTOS-EUROS atmospheric transport model.' Why not: 'The aim of this study is to determine the potential benefit of such a methodology to estimate the NH3 budget. Space-born observations from the Infrared Atmospheric Sounding Interferometer (IASI) and the LOTOS-EUROS atmospheric transport model are used.', or something on that line."

AC1.5: As suggested by the reviewer, the authors carefully re-read and edited the text in the entire manuscript. An English translator proofread and helped with editing the manuscript. The sentences that were changed are marked green in 'Manuscript changes'.

The authors appreciate the helpful comments of the reviewer.

**Anonymous Referee #2**

RC2.1: "The paper investigates the effects of using satellite-derived NH3 levels in a chemistry transport model on the modeled NH3 concentrations and deposition fluxes. The paper is interesting and easy to follow. I am in favor of its publication in ACP provided it address the points below."
AC2.1: The authors greatly appreciate the helpful and encouraging comments of the Referee.

RC2.2: "General Comments
- Can the authors elaborate on why results are different in the two years? - Is meteorology playing a role here? Is it possible to validate the meteorology to enrich the discussions?"
AC 2.2: The authors do believe that meteorology is playing an important role in the inter-annual differences in our results, as it influences both the satellite retrieval and (to a lesser extent) the model results. We looked at the meteorological circumstances in 2013-2014 have added the following section at the end of section 4.2.3.:

*"The inter-annual variations of the modelled and IASI-derived flux differences (see Figure 13 and 15) could be related to different meteorological conditions. The annual global climate reports from the NOAA (National Oceanic and Atmosphere Administration) show that the mean temperatures in Europe were higher in 2014 than in 2013, especially in western Europe. This might have had an effect on the emissions, which is only limited taken into account by the model. The annual precipitation in both years was near average for Europe as a whole. However, if we zoom in to a more regional scale, we see that it was much wetter than average during the warm season in nearly all parts of the Balkan Peninsula and Turkey (NOAA, 2014, 2015). Figure 13 shows that the largest inter-annual variations on a European scale occur around the Black Sea: in Ukraine, but also in the eastern parts of the Balkan Peninsula and Turkey. Some of these regions thus coincide with regions that experienced heavy rainfall in 2014 and might have affected emission and deposition processes which are not taken into account by the model. This suggests that meteorological effects might indeed influence our results. However, the examined period of two warm seasons only is too short to draw a conclusion."*

RC2.3:"- In addition, both the original and IASI inferred NH3 concentrations are overestimated both years. Can the authors discus why? Is it overestimation in emissions or underestimation in deposition?"
AC2.3: The modelled and the IASI-derived NH$_3$ concentrations are indeed overestimated in emission areas. We added the following section to '4.1.3. Summary of the comparison with in-site measurements' to discuss this:

*"In general, both the modelled and the IASI-derived concentrations seem to be overestimated in emission areas. This could potentially be related to the overpass time of the satellite. In high emission areas, the NH3 concentrations are more variable in time, and the IASI observations might have an uncertain representativeness. Moreover, the measurements in high emission areas are generally more uncertain with regard to their spatial representativeness. Overall, these measurements can be more affected by local rather than regional sources. Generally, the modelled and the observed NH3 total columns match quite well. This means that the LOTOS-EUROS model represents the spatial distribution of NH3 rather well. There are some areas with large discrepancies between the two where we see considerable deviations in the modelled and the IASI-derived concentrations. Most of these areas, however, cannot be validated against measurements, because of the lack of measurements here. The changes in the comparison of the available measurements with modelled versus IASI-derived concentrations are*

*therefore relativity small. Based on the measurements we have, we conclude that we do not see any significant improvement in the IASI-derived concentrations compared to the modelled concentrations."*

From our results, it is impossible to tell whether the differences between the two fields are related to a systematic or significant deviation in either the emissions or the deposition. There are so many different uncertain variables involved in both the model and the measurements that it is impossible to pinpoint the most important reason. This would be a very interesting, and challenging topic for potential follow-up studies.

RC2.4: "- Why are the deposition fluxes not evaluated against observations?"
AC2.4: The authors would very much like to evaluate the model against observations of dry deposition fluxes against observations. However, the available $NH_3$ dry deposition measurements in 2013 and 2014 are too limited to do a sensible model evaluation. There is certainly a need for more dry deposition measurements.

RC2.5: "Technical comments
Page 1 Line 33: : : :do not show strong improvements: : :.
Page 2, Line 30: : : :ALLOW us to : : :: : :
Section 2.2. needs some more explanation of how the uncertainty is calculated.
Section 2.4.1. needs more information on the temporal variation of emissions, in particular NH3.
Page 8, Line 16: Erisman (1993) estimated: : :.
Page 9, Line 16: : : :dry deposition fluxes IN Eq. (3):"
AC2.5: Thank you for the technical comments. We added some additional explanation about the IASI uncertainty to section 2.2:

*"The uncertainty estimate for each retrieved $NH_3$ total column is an error propagation of the individual parameter uncertainties. Whitburn et al. (2016) showed in an error characterization that individual retrieved $NH_3$ columns hold the smallest errors (~25%) in the situation of a high $NH_3$ concentration combined with a high thermal contrast. The error increases progressively when either of these lowers. In the case of a low $NH_3$ concentration and a low thermal contrast, the errors can be as high as ~270%."*

We also added a short section about the temporal variations of the emissions in LOTOS-EUROS to section 2.4.1:

[revised manuscript text omitted]